# CausalTab: Pretraining Across Causal Environments for Tabular Causal Discovery

## Abstract

Tabular causal discovery seeks directed structure from observational and optionally interventional samples. Causal discovery foundation models amortize this problem by mapping a dataset—values plus optional intervention masks—to graph hypotheses in one forward pass, yet often trail strong classical methods when pretraining coverage is narrow. We introduce CausalTab, a structured-tabular model with axial attention over variables and samples and asymmetric edge scoring, trained under *broad causal pretraining* over diverse graph priors, mechanisms, noise laws, dimensionalities, sample budgets, and observational versus mixed-interventional layouts, composed dynamically into episodes. We evaluate on seven synthetic benchmark families and a complementary semantic SCM benchmark with authored domains (still simulator-grounded). CausalTab achieves strong aggregate recovery with the largest gains when interventional evidence is available; a qualitative PCA on an administrative-workflow scenario suggests embeddings partially separate interpretable workflow roles, complementing synthetic scores. Together, these results highlight environment-rich pretraining as a practical ingredient for amortized causal discovery.

## 1. Introduction

Many scientific and operational decisions rest on causal structure, but data often arrive as tables: variables as columns and observational or interventional draws as rows (Peters et al., 2017). Classical causal discovery—constraint-based, score-based, or asymmetry-driven (Spirtes et al., 2000; Chickering, 2002; Hoyer et al.,

[1]Anonymous Institution, Anonymous City, Anonymous Region, Anonymous Country. Correspondence to: Anonymous Author <anon.email@domain.com>.

Preliminary work. Under review by the International Conference on Machine Learning (ICML). Do not distribute.

2008)—remains powerful when assumptions match the data-generating process, yet frequently requires per-dataset search, tuning, or large samples (Glymour et al., 2019; Uhler et al., 2013).

**Causal discovery foundation models** (CDFMs) pursue a different contract: amortize structure learning by mapping an *entire* dataset to a graph hypothesis in one forward pass (Montagna et al., 2024; Lorch et al., 2022). Tabular prediction foundations (Hollmann et al., 2023; 2025) target supervised labels from feature tables, whereas CDFMs instead emit directed graphs over variables with orientation and intervention layouts as first-class evidence (Lorch et al., 2022). We argue that progress on CDFMs is gated less by minor architectural tweaks than by **pretraining environment diversity**: narrow simulators invite dataset-specific shortcuts; transferable structure cues instead require exposure to heterogeneous causal *worlds*—varied graphs, mechanisms, noise laws, scales, and critically *observational versus mixed-interventional* layouts—composed into rich training tasks rather than a single fixed simulator recipe.

**CausalTab** instantiates this view with an axial-attention tabular encoder trained under **broad causal pretraining**: episodes sampled from a wide joint over graph priors, structural mechanisms, noise families, dimensionalities, sample budgets, and regime types, with dynamic composition so intervention patterns cannot substitute for genuine structural evidence. Experiments deliberately separate two axes: a generator-diverse *synthetic* benchmark that stresses anonymous mechanisms and obs vs. mixed-interventional coverage, and a *semantic* benchmark where SCMs retain simulator-known graphs but expose authored variable semantics for interpretable qualitative analysis alongside aggregate scores (still not observational field data). Together they test whether environment-rich pretraining improves amortized discovery under intervention-rich evidence while remaining competitive—and honest about ambiguity—when only passive observations are available.

**Contributions.** We foreground environment breadth as a first-order design axis for CDFMs; present CausalTab with an axial-attention tabular encoder and asymmetric edge scoring trained under a broad causal pretraining loop (Section 3); and show that synthetic and semantic aggregates

improve most reliably under mixed interventions. A semantic PCA probe further relates embeddings to workflow roles rather than treating columns as interchangeable anonymous synthetic coordinates.

## 2. Preliminaries

**Problem setup.** Causal discovery aims to recover the direct causal relations among $d$ features from data, typically represented by a directed acyclic graph $G = (V, E)$ with adjacency $A \in \{0, 1\}^{d \times d}$, where $A_{ij} = 1$ indicates a directed edge $i \rightarrow j$ (and $A_{ii} = 0$). At the mechanism level we consider structural causal models (SCMs) (Pearl, 2009): each feature $X_j$ is generated from its parents $\mathrm{pa}(j) := \{i \in V : i \rightarrow j \in E\}$ and exogenous noise. Observational data form a table $\mathbf{X} \in \mathbb{R}^{N \times d}$ whose rows are samples. We also consider mixed observational–interventional inputs: for sample $n$ and feature $j$, let $m_j^{(n)} \in \{0, 1\}$ mark whether $X_j$ was intervened when drawing row $n$, with $\mathbf{m}^{(n)} = \mathbf{0}$ for purely observational rows. The task is to estimate the underlying structure (e.g., $A$ or edge-direction probabilities) from these inputs.

**Evaluation and amortized inference.** On simulator-known graphs we evaluate predicted directed edges against ground truth using directed edge-level $F_1$ and structural distances SHD and SID (Peters & Bühlmann, 2015). A causal discovery foundation model uses a shared predictor across datasets with varying sample sizes $N$, feature counts $d$, data-generating mechanisms, and optional intervention masks, amortizing structure inference in a single forward pass rather than per-dataset conditional-independence testing, combinatorial search, or continuous optimization; the scale and diversity of causal environments encountered in training—graph priors, mechanisms, noise, dimensions, sample budgets, and intervention regimes—shape what structural cues can transfer to unseen datasets.

## 3. Method

CausalTab combines an amortized tabular encoder with **broad causal pretraining**: the encoder maps an entire dataset—values plus optional intervention indicators—to directed edge probabilities in *one forward pass*; pretraining exposes the model to many distinct causal *environments* so test-time generators are less likely to fall outside the training support.

**Inputs and outputs.** Columns index variables $1, \ldots, d$ and rows index observational or interventional draws (Peters et al., 2017). Observed values $\mathbf{X}^{\mathrm{val}} \in \mathbb{R}^{N \times d}$ are paired with binary masks $\mathbf{M} \in \{0, 1\}^{N \times d}$ marking structurally intervened coordinates on each row (purely observational rows use $\mathbf{M} = \mathbf{0}$). Following Lorch et al. (2022), we stack

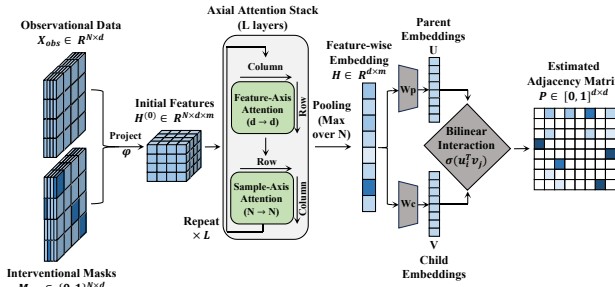

*Figure 1.* **Model architecture overview.** Embedded value–mask pairs pass through alternating axial attention blocks (variables vs. samples) before pooling to variable tokens for asymmetric directed edge scoring.

channels into $\mathbf{X} \in \mathbb{R}^{N \times d \times 2}$. One forward pass yields $\widehat{\mathbf{P}} \in (0, 1)^{d \times d}$, optionally thresholded to an adjacency. Benchmarks that expose simulator-known graphs use edge-level $F_1$, SHD, and SID (Peters & Bühlmann, 2015).

**Axial-attention encoder.** Each $(x_{nj}, m_{nj})$ embeds with a shared linear map. Transformer blocks then *alternate* multi-head self-attention over the feature axis and over the sample axis (Vaswani et al., 2017; Ho et al., 2019): along variables we mix across columns at fixed sample index; along samples we mix across rows at fixed column index. Each block uses residual connections and feed-forward layers around these axial attention updates, as detailed in Appendix B. After $L$ alternations, max-pooling over $N$ produces one token $\widetilde{\mathbf{h}}_j$ per variable (Zaheer et al., 2017). Figure 1 depicts this pipeline.

**Asymmetric directed edge head.** Pairs of variable tokens map through learned parent and child projections into a shared space; normalized similarity scores with learned calibration yield $\hat{p}_{ij}$ for $i \rightarrow j$ without an outer combinatorial search over DAGs (Dozat & Manning, 2017; Lorch et al., 2022). Layer equations appear in Appendix B; training hyperparameters and optimizer choices are documented with the architecture specification in Appendix F.

**Broad causal pretraining.** The substantive modeling decision is the *distribution of causal worlds* encountered during training. Every episode samples a graph prior, structural-equation family, noise law, dimension $d$, counts of observational versus interventional rows, and (when needed) a mixed-interventional mask pattern; an SCM simulator emits tensors consistent with Section 3 together with known adjacency $A^\star$. Pretraining minimizes the expected directed edge-wise binary cross-entropy between $A^\star$ and $\widehat{\mathbf{P}} = f_\theta(\mathbf{X})$, averaged over off-diagonal ordered pairs $(i, j)$ with $i \neq j$, as formalized in Appendix B. Episodes *resample* these ingredients rather than cycling a single benchmark recipe: graph ensembles, mechanism idioms, noise types,

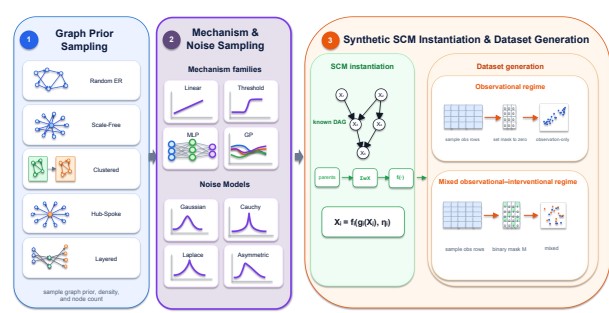

*Figure 2.* **Broad causal pretraining loop.** Sample worlds (graphs, mechanisms, noise, $(d, N)$, regimes), simulate SCMs with known DAGs, supervise edge prediction; Appendix F lists implementation details.

$(d, N)$ pairs, and intervention layouts vary so learners cannot exploit fixed layout shortcuts. The aim is **broader task support across causal environments**—not merely more rows from one simulator—matching the diversity of our anonymous synthetic and semantic evaluations. Training omits differentiable acyclicity constraints; optional cycle pruning runs only at inference. Concrete instantiations of priors $\mathcal{G}$, mechanisms $\mathcal{F}$, noise laws $\mathcal{P}_\epsilon$, dimensions $\mathcal{D}$, sample budgets $\mathcal{N}$, and the joint resampling rule over episodes are listed in Appendix F.3; Appendix H.1 ablates environment breadth.

Figure 2 summarizes the training-time episode generator.

## 4. Experiments

We evaluate along two complementary axes that isolate different evaluation goals.

**Synthetic benchmark (anonymous breadth).** Seven anonymous mechanism families—`linear_gauss`, `linear_nongauss`, `linear_graph`, `gp_simple`, `gp_hard`, `mul_noise`, and `pfn`—designed to span linear, smooth nonlinear, highly oscillatory, heteroscedastic, and broader nonlinear structural equations. For each family we sweep graph sizes $d \in \{5, 10, 20\}$ and two evidence regimes: **observation-only** with $N{=}1000$ observational rows, and **mixed-interventional** data with 800 observational rows plus 200 single-variable structural interventions allocated as evenly as possible across variables. Every (family, regime, $d$) configuration uses 100 independent instances with fixed random seeds (Lorch et al., 2022; Zheng et al., 2018). This benchmark stress-tests *broad robustness*: many generators and both observational and interventional layouts, without interpretable variable names.

**Semantic benchmark (domain-grounded probe).** Anonymous synthetic benchmarks provide simulator-known graphs but weak variable-level semantics, which limits interpretable failure analysis. We evaluate on a *domain-grounded semantic causal environment benchmark* that remains **synthetic** and **SCM-controlled**: authored scenarios attach semantics to variables, directed edges, mechanism assignments, edge rationales, and intervention handles while preserving simulator-known DAGs for evaluation—not field observational telemetry. Alongside the anonymous families above, this suite is intended as a *complementary* out-of-distribution-style evaluation axis for interpretable qualitative analysis—including workflow-oriented probes below—rather than as a substitute for broad anonymous stress testing or for field observational studies. Each scenario specifies an SCM with domain-interpretable variables and directed mechanisms; structured authoring specifications compile into instantiated SCMs, and the generator emits observational and mixed-interventional tabular datasets with simulator-known adjacencies and the same value–mask layout as Section 3. Domain-interpretable names and narrative fields support scenario authoring and interpreting outcomes; methods are still evaluated on those value–mask tensors rather than on raw field telemetry. The suite spans **100** scenarios across **10** thematic domains (e.g., government services, healthcare-style workflows, logistics-style processes) under observation-only and mixed-interventional regimes, yielding **200** dataset–regime tasks; aggregate metrics measure transfer to authored semantics, and named variables enable qualitative probes (below). Appendix C provides the benchmark inventory, schema, sampling and generation rules, validation checks, and audit protocol; a structured LLM-aided pass supports documentation consistency only and is not exhaustive semantic certification.

**Baselines and metrics.** We compare thirteen methods spanning amortized inference, continuous-optimization structure learners, constraint-based classical algorithms, and regression-based controls (including AVICI, NOTEARS / NOTEARS-MLP, IGSP, CDIS, GIES, LiNGAM, DCDI, NoDAGS, PC, DAS, and Random-Regress). Some methods apply only to observation-only or only to mixed-interventional layouts; macro summaries mark inapplicable cells with "–". All runs optimize the same edge-level metrics against simulator-known graphs: $F_1$ (higher better), SHD (lower better), and SID (lower better), using the shared evaluation protocol documented in Appendix I.

**Aggregate results.** Table 1 lists representative methods with macro-means on synthetic data (equal weight over all family$\times d$ configurations) and regime means on semantic

*Table 1.* **Representative synthetic vs. semantic aggregates.** Synthetic: macro-mean rank ($\downarrow$) and $F_1$ ($\uparrow$) over seven families and $d \in \{5, 10, 20\}$ under observation-only (1000 samples) and mixed-interventional (800+200) regimes. Semantic: regime-mean $F_1$ ($\uparrow$) over 200 dataset–regime tasks. "–" denotes a regime-inapplicable method. Full SHD/SID and all thirteen methods: Tables 9 and 2.

| Method | Syn. obs. | | Syn. mixed | | Semantic | |
|---|---|---|---|---|---|---|
| | Rnk$\downarrow$ | F1$\uparrow$ | Rnk$\downarrow$ | F1$\uparrow$ | Obs. F1$\uparrow$ | Mix. F1$\uparrow$ |
| RandomRegress | 11.6 | 0.240 | – | – | 0.206 | – |
| DAS | 3.4 | 0.699 | – | – | 0.242 | – |
| PC | 6.9 | 0.496 | – | – | 0.649 | – |
| GIES | 5.0 | 0.599 | 2.6 | 0.780 | 0.698 | 0.882 |
| AVICI | 6.0 | 0.551 | 2.7 | 0.844 | 0.560 | 0.943 |
| DCDI | 8.3 | 0.430 | 5.9 | 0.459 | 0.338 | 0.435 |
| NoDAGS | – | – | 4.7 | 0.573 | – | 0.864 |
| CausalTab | 1.9 | 0.824 | 1.0 | 0.952 | 0.681 | 0.971 |

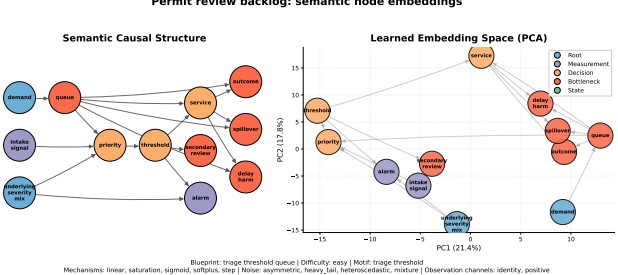

**Permit review backlog: semantic node embeddings**

*Figure 3.* **Semantic qualitative probe.** *Permit review backlog* SCM (*left*) and PCA of embeddings (*right*); interpret as workflow roles, not literal DAG recovery.

tasks; Tables 9 and 2 contain full SHD/SID columns and the complete baseline pool.

On synthetic data, CausalTab achieves the best average rank in both observation-only and mixed-interventional regimes and attains the strongest aggregate mixed-interventional $F_1$ among applicable methods, with competitive SHD/SID in the same regime. Observation-only synthetic rankings remain tighter: specialized classical or continuous-optimization methods occasionally win on individual families when their assumptions align with the generator, underscoring that observational evidence alone is intrinsically more ambiguous. Semantic aggregates echo this pattern: under mixed interventions CausalTab achieves the strongest reported $F_1$ among applicable methods, whereas observation-only semantic scores remain more overlapping across methods—consistent with viewing the semantic suite as an interpretable OOD-style axis rather than an easier substitute for the synthetic stress test. Figure 10 in the appendix visualizes per-configuration $F_1$ across dataset–regime columns for the full baseline pool.

**Qualitative semantic embedding (*Permit review backlog*).** To connect representation geometry to authored semantics, Figure 3 shows an observational draw from the *Permit review backlog* scenario (government services; `triage_threshold_queue` blueprint). The left panel plots the simulator-known SCM; the right panel shows a PCA of final-layer node embeddings $\widetilde{\mathbf{h}}_i$. PCA is only a qualitative lens—axes are not causal directions and distances are not faithful DAG geometry—but the projection separates coarse **workflow roles**: upstream demand/background quantities drift away from a queue-centered consequence region (queue pressure, spillover, delays), triage and threshold decisions lie between those poles, and service progression sits somewhat apart as the transition from selection to active processing. We highlight modest functional separation rather than tight clustering or structural isomorphism. Appendix D reports additional scaling analyses on observational data, including sample-size curves with $N \in \{10, 100, 1000, 10000\}$ and higher-$d$ evaluations at $d \in \{50, 100, 300\}$ among methods that complete within the evaluation budget (some runs omit larger $N$ or $d$ due to timeouts).

## 5. Conclusion and Limitations

**CausalTab** ties together two ideas emphasized throughout this workshop paper: *amortized* tabular causal discovery should be trained like other foundation models—not on a single synthetic niche—and *broad causal pretraining* supplies the diversity of graphs, mechanisms, noise laws, dimensionalities, sample budgets, and observational versus mixed-interventional layouts needed for that goal. Empirically, macro summaries on anonymous synthetic families and semantic SCM environments consistently reward CausalTab when interventions are present, while observation-only rows remain more contested—a pattern we expect whenever passive samples underdetermine directionality.

The semantic suite adds value beyond scalar rankings: qualitative probes such as Figure 3 connect representation geometry to workflow language, reinforcing that simulator-grounded semantics can expose how encoders organize variables even though PCA axes are not causal coordinates.

**Limitations.** Evidence remains synthetic or authored-semantic rather than field-validated. Observation-only ambiguity, scaling to very large $d$, hidden confounding, cycles, temporal structure, selection bias, and calibrated uncertainty are important open directions for future amortized causal discovery systems.

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

# A. Extended Preliminaries

## A.1. Causal discovery: inputs, outputs, and evaluation

Causal discovery aims to recover the direct causal relations among a set of features from data, typically represented as a directed acyclic graph (DAG). Let the feature index set be $V := \{1, \ldots, d\}$ and the causal graph be $G = (V, E)$ with $E \subseteq V \times V$. We use an adjacency matrix $A \in \{0, 1\}^{d \times d}$ to represent $G$, where $A_{ij} = 1$ indicates a directed edge $i \to j$ (and $A_{ii} = 0$).

At the mechanism level, we consider a structural causal model (SCM) (Pearl, 2009): each feature $X_j$ is generated from its parents $\mathrm{pa}(j) := \{i \in V : i \to j \in E\}$ via

$$X_j = f_j\big(X_{\mathrm{pa}(j)}, \epsilon_j\big), \qquad j = 1, \ldots, d, \qquad (1)$$

where $\epsilon_j$ is a noise term. Observational data are given as a table (data matrix) $\mathbf{X} \in \mathbb{R}^{N \times d}$, where each row is one sample:

$$\mathcal{D}^{\mathrm{obs}} = \{\mathbf{x}^{(n)}\}_{n=1}^N, \qquad \mathbf{x}^{(n)} \in \mathbb{R}^d. \qquad (2)$$

We also consider mixed observational–interventional inputs, where each sample is accompanied by an intervention indicator vector. Concretely, for sample $n$ and feature $j$, let $m_j^{(n)} \in \{0, 1\}$ denote whether $X_j$ is intervened on when generating $\mathbf{x}^{(n)}$. The input can be written as

$$\mathcal{D} = \{(\mathbf{x}^{(n)}, \mathbf{m}^{(n)})\}_{n=1}^N, \qquad \mathbf{m}^{(n)} \in \{0, 1\}^d, \qquad (3)$$

where $\mathbf{m}^{(n)} = \mathbf{0}$ for purely observational samples. The causal discovery task is to estimate the underlying structure (e.g., $A$ or edge-direction probabilities) from $\mathcal{D}$. We evaluate the predicted directed edge set $\widehat{E}$ against the ground-truth $E^\star$ using directed edge-level $F_1$ as well as structural distances, specifically Structural Hamming Distance (SHD) and Structural Intervention Distance (SID) (Peters & Bühlmann, 2015).

## A.2. Foundation-model viewpoint: amortized dataset-to-graph inference

A causal discovery *foundation model* reframes structure learning as a shared, reusable mapping from datasets to graphs. Formally, let $\mathfrak{D}$ denote the space of possible inputs, including varying sample sizes $N$, feature counts $d$, and optional intervention masks. A foundation model is a parameterized predictor

$$f_\theta : \mathfrak{D} \to \mathfrak{Y}, \qquad \widehat{Y} = f_\theta(\mathcal{D}), \qquad (4)$$

where the output space $\mathfrak{Y}$ can be an edge-score/probability matrix or a discrete adjacency estimate.

A key feature of the foundation-model formulation is adaptiveness across heterogeneous causal-discovery settings without changing the inference procedure: the same predictor $f_\theta$ can be applied to datasets with varying sample sizes $N$, feature counts $d$, and data-generating mechanisms, while optionally leveraging interventional signals when available. Crucially, inference is amortized: instead of performing CI testing, combinatorial search, or continuous optimization separately for every dataset, one applies the same $f_\theta$ to obtain structure predictions in a single forward pass. From this viewpoint, the training distribution is part of the model design. A CDFM is expected to acquire reusable structural inductive biases from many pretraining tasks, so the scale and diversity of causal environments—including graph priors, mechanisms, noise distributions, dimensions, sample sizes, and intervention regimes—directly affect what structural cues can transfer to unseen datasets.

# B. Method Details and Equations

This section restates the architecture and pretraining objective with full notation; it parallels the main-text summary.

## B.1. Architecture equations

A shared linear map $\phi$ embeds each value–indicator pair into width $m$,

$$\mathbf{H}^{(0)} = \phi(\mathbf{X}) \in \mathbb{R}^{N \times d \times m}. \qquad (5)$$

The encoder stacks $L$ blocks that alternate multi-head self-attention over the feature axis and over the sample axis (with residuals and feed-forward layers), capturing cross-feature structure within rows and cross-sample regularities within columns (Vaswani et al., 2017; Ho et al., 2019). Schematically,

$$\mathbf{H}^{(\ell)} = \mathrm{Attn}_{\mathrm{samp}}\Big(\mathrm{Attn}_{\mathrm{var}}\big(\mathbf{H}^{(\ell-1)}\big)\Big), \quad \ell = 1, \ldots, L. \qquad (6)$$

We max-pool over samples to obtain feature tokens $\widetilde{\mathbf{H}} \in \mathbb{R}^{d \times m}$ (Zaheer et al., 2017), apply learnable parent and child projections $\mathbf{W}_{\mathrm{p}}, \mathbf{W}_{\mathrm{c}}$,

$$\mathbf{U} = \widetilde{\mathbf{H}}\mathbf{W}_{\mathrm{p}}, \qquad \mathbf{V} = \widetilde{\mathbf{H}}\mathbf{W}_{\mathrm{c}}, \qquad (7)$$

and score each ordered pair $(i, j)$ with normalized cosine similarity, learnable temperature $\tau$, and bias $b$ (Dozat & Manning, 2017; Lorch et al., 2022),

$$\widehat{S}_{ij} = e^\tau \left\langle \frac{\mathbf{u}_i}{\|\mathbf{u}_i\|_2}, \frac{\mathbf{v}_j}{\|\mathbf{v}_j\|_2} \right\rangle + b, \quad \widehat{P}_{ij} = \sigma(\widehat{S}_{ij}),$$
$$\widehat{A}_{ij} = \mathbb{1}[\widehat{P}_{ij} \geq \tau_{\mathrm{thr}}] \ (i \neq j), \ \widehat{A}_{ii} = 0. \qquad (8)$$

Here $\tau_{\mathrm{thr}}$ is a decoding threshold (distinct from $\tau$).

## B.2. Pretraining episode and objective

Let $\mathcal{G}$, $\mathcal{F}$, and $\mathcal{P}_\epsilon$ denote catalogs of graph priors, structural-equation mechanism families, and exogenous

noise models, respectively; let $\mathcal{D}$ and $\mathcal{N}$ denote admissible feature dimensions $d$ and dataset sizes $N$; and let $\mathcal{R} = \{\mathrm{obs}, \mathrm{mix}\}$ encode observational versus mixed observational–interventional regimes. Each pretraining episode draws a configuration

$$z = (g, f, \rho, d, N, r) \sim \mathcal{S},$$
$$g \in \mathcal{G}, \ f \in \mathcal{F}, \ \rho \in \mathcal{P}_\epsilon, \ d \in \mathcal{D}, \ N \in \mathcal{N}, \ r \in \mathcal{R}, \quad (9)$$

where $\mathcal{S}$ denotes the engine's joint resampling rule over these factors (we leave marginals abstract here). Given $z$, the simulator constructs an SCM with simulator-known adjacency $A^\star$,

$$(A^\star, \mathcal{M}_z) = \mathrm{SCM}(z), \quad (10)$$

and generates observations and binary intervention indicators consistent with regime $r$ and sample budget $N$,

$$(\mathbf{X}^{\mathrm{val}}, \mathbf{M}) \sim \mathrm{Gen}(\mathcal{M}_z, r, N),$$
$$\mathbf{X} = [\mathbf{X}^{\mathrm{val}}, \mathbf{M}] \in \mathbb{R}^{N \times d \times 2}. \quad (11)$$

The supervised pretraining tuple is $\upsilon = (\mathbf{X}, A^\star)$. Concrete instantiations of $\mathcal{G}, \mathcal{F}, \mathcal{P}_\epsilon, \mathcal{D}, \mathcal{N}$, and $\mathcal{S}$ are listed in Appendix F.3; Appendix H.1 ablates environment breadth.

When $r = \mathrm{obs}$, $\mathbf{M}$ is all-zero; when $r = \mathrm{mix}$, $\mathbf{M}$ marks intervened coordinates on interventional rows while observational rows remain zero, following the layout in Section 3.

Pretraining minimizes the expected directed edge-wise BCE over tasks sampled by this engine: letting $\mathcal{P}_{\mathrm{train}}$ denote the distribution of $(\mathbf{X}, A^\star)$ induced by $z \sim \mathcal{S}$ followed by Gen, we minimize the average loss between $A^\star$ and $\widehat{\mathbf{P}} = f_\theta(\mathbf{X})$ over off-diagonal ordered pairs,

$$\min_\theta \mathbb{E}_{(\mathbf{X}, A^\star) \sim \mathcal{P}_{\mathrm{train}}} \big[ \mathcal{L}_{\mathrm{BCE}}(\theta) \big],$$
$$\mathcal{L}_{\mathrm{BCE}}(\theta) = \frac{1}{d(d-1)} \sum_{i \neq j} \big[ -A^\star_{ij} \log \widehat{P}_{ij}$$
$$- (1 - A^\star_{ij}) \log(1 - \widehat{P}_{ij}) \big].$$

At test time, we apply the same amortized map without dataset-specific fine-tuning and threshold $\widehat{\mathbf{P}}$ when a discrete graph is needed (Eq. (8)). Training uses no explicit differentiable acyclicity constraint; when strict DAG outputs are required at evaluation time, we optionally apply lightweight cycle pruning as inference-only decoding post-processing.

## C. Semantic Causal Environment Benchmark

### C.1. Benchmark design

Anonymous synthetic benchmarks provide simulator-known graphs but weak variable-level semantics, which limits interpretable failure analysis. Section 4 summarizes the domain-grounded semantic suite at a high level; here

we document inventory and generation rules. The benchmark remains **synthetic** and **SCM-controlled** rather than field observational telemetry. The generator compiles structured specifications into SCMs and emits observational and mixed-interventional tabular datasets (value tensors and binary masks aligned with Section 3). The suite complements the anonymous synthetic benchmark in Section 4 as an out-of-distribution-style evaluation axis; it spans **100** scenarios across **10** thematic domains under obs and int, yielding **200** dataset–regime tasks. Table 3 summarizes the benchmark inventory; schema, sampling and generation rules, validation checks, and audit protocol appear in Appendix G.2. A structured LLM-aided pass supports documentation consistency only and is not exhaustive semantic certification.

### C.2. Aggregate semantic benchmark results

Table 2 aggregates semantic performance by regime: CausalTab attains the strongest F1, SHD, and SID among applicable methods under mixed-interventional evidence, while observation-only settings stay more ambiguous and competitive across baselines; the benchmark complements anonymous synthetic evaluation but remains simulator-grounded rather than field deployment evidence, and rankings can depend on thresholds and runtime budgets.

## D. Analysis

### D.1. Sample-size scaling

Figure 4 plots F1 versus $N \in \{10, 100, 1000, 10000\}$ on observational gp_hard_obs and pfn_obs; some methods are omitted at larger $N$ due to timeouts or run failures. CausalTab generally improves with more samples, especially on gp_hard_obs, whereas pfn_obs stays relatively competitive across baselines; DAS and Random-Regress provide additional ordering/regression controls.

### D.2. Scalability to high dimensions

Figure 5 reports SHD for gp_hard_obs and pfn_obs at $d \in \{50, 100, 300\}$ among methods that complete under the evaluation budget; growing $d$ expands the directed-edge candidate set quadratically and many baselines time out or return incomplete results. Across displayed settings, CausalTab ranks among the strongest or near-strongest methods, while RandomRegress remains an ordering/regression control with substantially worse SHD.

### D.3. Semantic embedding visualization

The main paper presents the PCA visualization for the *Permit review backlog* semantic scenario and discusses how to interpret it as workflow-functional geometry rather than lit-

*Table 2.* **Semantic causal environment benchmark.** Aggregate performance reported separately for observation-only and mixed-interventional semantic regimes. Lower SHD and SID are better; higher F1 is better. Missing entries indicate methods not applicable to the regime.

| Method | Observation-only | | | Mixed-interventional | | |
| --- | --- | --- | --- | --- | --- | --- |
| | F1↑ | SHD↓ | SID↓ | F1↑ | SHD↓ | SID↓ |
| RandomRegress | 0.206 | 24.233 | 48.19 | – | – | – |
| DAS | 0.242 | 17.540 | 62.47 | – | – | – |
| LiNGAM | 0.303 | 11.280 | 41.06 | – | – | – |
| PC | 0.649 | 6.710 | 35.23 | – | – | – |
| CDIS | 0.595 | 7.780 | 38.09 | 0.604 | 7.520 | 37.02 |
| GIES | 0.698 | 7.340 | 27.63 | 0.882 | 3.860 | 3.95 |
| IGSP | 0.664 | 6.270 | 30.70 | 0.611 | 7.360 | 34.57 |
| NOTEARS | 0.203 | 11.920 | 43.58 | – | – | – |
| NOTEARS-MLP | 0.270 | 11.550 | 53.92 | – | – | – |
| NoDAGS | – | – | – | 0.864 | 3.195 | 13.97 |
| DCDI | 0.338 | 32.380 | 69.10 | 0.435 | 32.970 | 70.13 |
| AVICI | 0.560 | 8.300 | 34.54 | 0.943 | 1.700 | 6.39 |
| CausalTab | 0.681 | 6.071 | 27.45 | 0.971 | 0.878 | 1.89 |

eral DAG recovery (Figure 3). Additional quantitative analyses of embeddings, graph-statistic prediction probes, and wall-clock runtime comparisons on the synthetic benchmark appear in Appendix H.2 and Appendix H.3.

# E. Related Work

**Classical causal discovery methods**. Classical causal discovery methods are commonly grouped into three classes (Vowels et al., 2023): constraint-based approaches grounded in conditional-independence (CI) relations (Spirtes et al., 2000), score-based approaches that optimize a graph score over a structure space (Chickering, 2002), and functional-assumption approaches that identify directions under additional assumptions on the data-generating mechanisms (Hoyer et al., 2008). A representative constraint-based method is the PC algorithm, which first recovers an undirected skeleton via CI tests and then applies orientation rules to obtain a partially directed equivalence-class representation (Andersson et al., 1997). When interventional information is available, distribution shifts induced by interventions can further improve orientability and identifiability; this idea is formalized in methods such as IGSP (Wang et al., 2017). Score-based methods maximize decomposable scores via search; foundational greedy analyses appear in Chickering (Chickering, 2002), with GIES (Hauser & Bühlmann, 2012) extending score-based search to interventional settings. To reduce explicit search, another line of work formulates structure learning as differentiable or continuous optimization and learns an adjacency matrix directly with gradient-based methods. NOTEARS (Zheng et al., 2018) is a landmark example, introducing a differentiable acyclicity constraint combined with sparsity regularization, and it inspired variants with alternative objectives or constraints such as

NoDAGS (Sethuraman et al., 2023). While these methods can be effective in low-to-moderate dimensions when assumptions are well matched, they often struggle in high-dimensional and finite-sample regimes due to the heavy computational burden of repeated tests and the statistical fragility of CI or score estimation (Colombo & Maathuis, 2014; Uhler et al., 2013). Functional-assumption methods, such as LiNGAM (Shimizu et al., 2006), can identify directions under linear non-Gaussian independent noise, but may be sensitive to assumption mismatch, optimization details, and hyperparameter choices such as penalties and sparsification thresholds. Varsortability can also make simulated DAG benchmarks easier than intended, motivating ordering/regression control baselines (Reisach et al., 2021).

**Tabular foundation models and causal discovery foundation models**. Tabular foundation models, exemplified by TabPFN (Hollmann et al., 2023; 2025), show that large-scale pretraining over synthetic tabular tasks yields strong in-context predictors for supervised learning. They are primarily designed to predict labels or targets from feature tables, whereas tabular causal discovery outputs a directed graph over the variables themselves. Thus strong tabular prediction does not by itself solve the causal discovery problem, where edge orientation, intervention information, and structural ambiguity are central.

Recent amortized causal discovery maps observational and/or interventional datasets directly to edge probabilities or adjacencies in one forward pass (Montagna et al., 2024). AVICI (Lorch et al., 2022) trains such a predictor on synthetic graphs and mechanisms. We follow this CDFM paradigm but emphasize *causal pretraining task construction*: broader environments, heterogeneous composed tasks, and evaluation on anonymous synthetic benchmarks, semantic causal environments, and scaling analy-

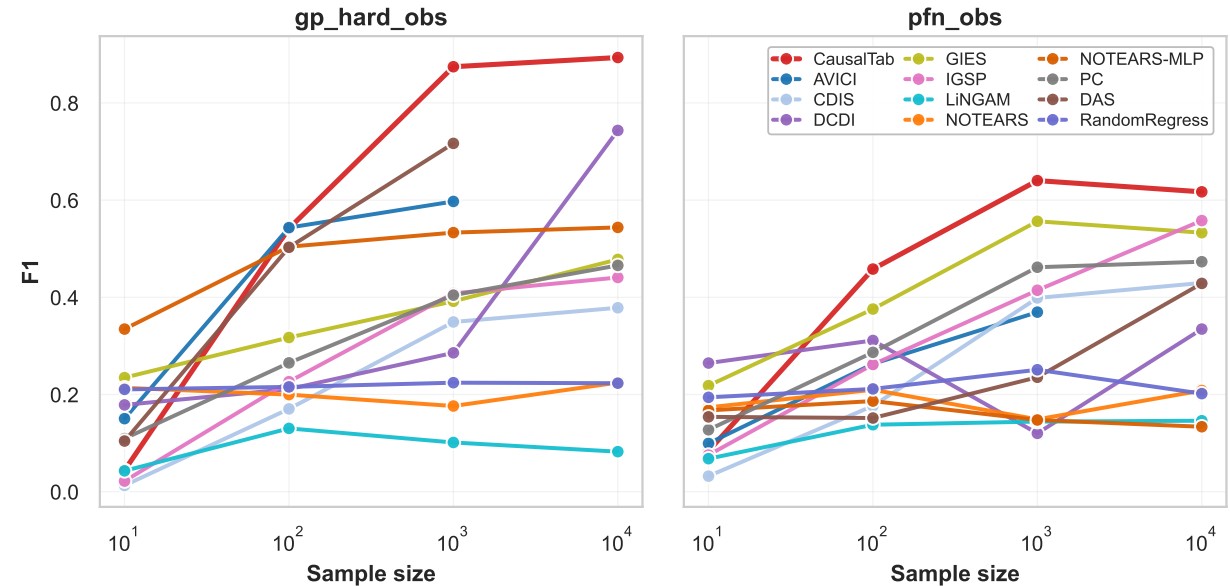

*Figure 4.* **Sample-size scaling on observational data.**

ses.

## F. Model Architecture and Training Details

### F.1. Model Architecture

The architecture schematic and broad causal pretraining loop appear as Figure 1 and Figure 2 in the main text; this section records implementation choices and training hyperparameters.

CausalTab employs a transformer-based encoder-decoder architecture optimized for causal structure learning from tabular data.

**Feature encoder.** The encoder processes input data through an axial attention mechanism that alternates between the observation axis and variable axis. Our architecture consists of 8 transformer blocks, with each block applying attention along one axis before transposing to the other axis, yielding 16 effective layers. Key architectural specifications:

- Hidden dimension: 128
- Multi-head attention: 8 heads with 32-dimensional keys/queries per head
- Feed-forward network: Expansion factor of 4 (hidden size 512)
- Regularization: 0.1 dropout rate
- Aggregation: Max pooling along the observation dimension

The axial design allows the model to capture both cross-variable dependencies and cross-sample distributional pat-

terns efficiently. Given input $\mathbf{X} \in \mathbb{R}^{B \times N \times d \times 2}$ (batch, observations, variables, channels), the encoder alternates attention operations along dimensions $N$ and $d$, enabling scalability to graphs with hundreds of variables.

**Graph prediction head.** Edge probabilities are computed via normalized similarity between asymmetric projections of node representations. Specifically, for node embeddings $\mathbf{h}_i$, we learn two projection functions $f_u(\cdot)$ and $f_v(\cdot)$ and compute:

$$\mathbf{P}_{ij} = \exp(\tau) \cdot \frac{f_u(\mathbf{h}_i)^\top f_v(\mathbf{h}_j)}{\|f_u(\mathbf{h}_i)\|\|f_v(\mathbf{h}_j)\|} + b, \qquad (12)$$

where $\tau$ is a learnable temperature (initialized at 2.0) and $b$ is a learnable bias (initialized at -1.0). The L2 normalization bounds the similarity scores and improves training stability.

### F.2. Training Config

**Optimizer configuration.** We employ the LAMB optimizer (Layer-wise Adaptive Moments optimizer for Batch training) with adaptive learning rate scheduling. The base learning rate $\eta_{\text{base}} = 3 \times 10^{-5}$ is scaled according to the effective batch size:

$$\eta = \eta_{\text{base}} \cdot \sqrt{B_{\text{eff}}}, \qquad (13)$$

where $B_{\text{eff}}$ denotes the maximum effective batch size observed during training. This square-root scaling maintains optimization dynamics consistent across different batch sizes. We use no weight decay, apply gradient clipping at

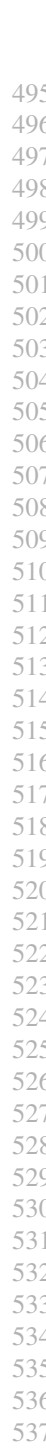

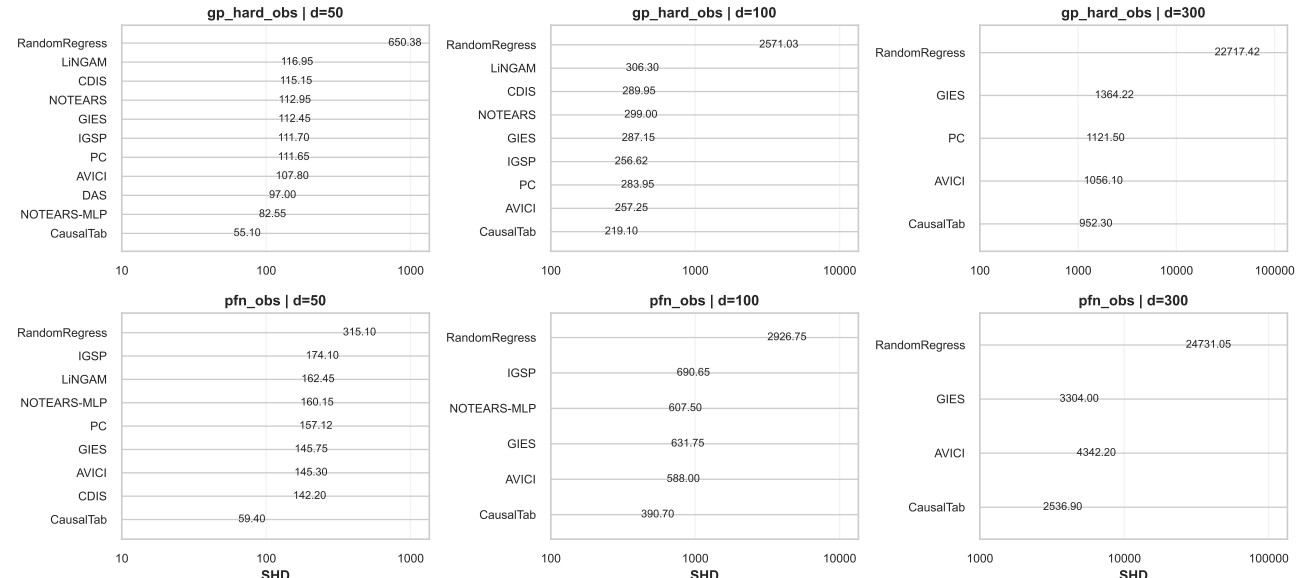

*Figure 5.* **High-dimensional observational scalability.** SHD (lower is better; log-scale axis) for `gp_hard_obs` and `pfn_obs` at $d \in \{50, 100, 300\}$.

magnitude 1.0, and maintain a constant base learning rate schedule for the duration of pretraining (scaled by effective batch size as above).

**Mixed-precision training.** Training is conducted entirely in BFloat16 arithmetic without loss scaling. BFloat16 provides greater dynamic range than FP16, eliminating the need for gradient scaling and reducing numerical instability. We leverage hardware-accelerated mixed precision through PyTorch's autocast mechanism.

**Adaptive batch scheduling.** To balance memory constraints and sample efficiency across varying graph dimensions, we use a dimension-dependent *per-device micro-batch* size. For graph dimension $d$, the micro-batch size on each accelerator follows

$$B_d = \left\lfloor 0.1 + \frac{1000 \cdot (1 - e^{-0.1d})}{d} \right\rfloor. \quad (14)$$

This damped schedule assigns larger micro-batches to small graphs (memory-efficient) and smaller micro-batches to large graphs (memory-intensive). Each optimizer update accumulates gradients over 10 consecutive micro-batches before stepping.

**Dimension-balanced sampling.** Dimensions are sampled with probability weights inversely related to their effective throughput under the micro-batch schedule above, accounting for the number of accelerator devices and the fixed gradient-accumulation depth (10 micro-batches per

optimizer step). Concretely, we use weights of the form

$$p(d) \propto \frac{d}{B_d \cdot N_{\text{GPU}}}, \quad (15)$$

which encourages comparable exposure across dimensions despite heterogeneous per-step costs.

**Data augmentation via intervention mixing.** Training alternates between observational-only and mixed observational–interventional episodes drawn from the same environment distribution. With probability 0.5, an episode is observational-only ($n_{\text{int}} = 0$); otherwise it is a mixed-regime episode. In mixed episodes, the interventional sample count $n_{\text{int}}$ is drawn approximately log-uniformly from an interval with lower endpoint $\max(d, 1)$ and upper endpoint 200, where $d$ is the graph dimension. Conditional on $n_{\text{int}}$, the observational sample count $n_{\text{obs}}$ is drawn approximately log-uniformly from an interval with lower endpoint $\max(100, 2n_{\text{int}})$ and upper endpoint 1000. Sampled counts are then clipped by configured episode caps; for the final training setup these caps correspond to at most 600 observational and 200 interventional samples per episode.

**Computational requirements.** Pretraining was run on $2\times$ NVIDIA RTX 4090 GPUs using BFloat16 mixed precision as described above. The full training run required on the order of ten days of wall-clock time on this hardware.

## F.3. Pretraining Data Configuration

A critical component of CausalTab is its pretraining phase, where the model learns reusable dataset-to-graph inference behavior from a broad distribution of causal environments before deployment. Unlike AVICI, which trains on a narrower simulator with limited mechanism coverage (primarily linear and smooth GP functions), CausalTab explores causal pretraining task construction with varying environment coverage to systematically investigate the impact of simulator diversity on out-of-distribution generalization. This design choice reflects our hypothesis that richer causal environments and more varied task compositions enable stronger transfer to unseen causal structures—a claim we validate through a prior-strength ablation (Section H.1).

We consider two pretraining configurations that differ substantially in their coverage of causal mechanisms and graph structures: the **weak-prior** variant focuses on well-studied mechanism families similar to prior work, while the **strong-prior** variant significantly expands the distribution to include challenging nonlinear, non-additive, and heteroscedastic mechanisms.

**Weak-prior pretraining.** This configuration serves as our baseline and covers fundamental causal mechanisms:

- **Graph structures**: Erdős-Rényi (edges per variable: 1.0, 2.0, 3.0), Scale-Free and its transpose (edges: 1–3, power: 0.7/1.0/1.2/1.5), Watts-Strogatz (dimension: 2/3, neighbors: 1/2, rewiring: 0.2/0.4), Stochastic Block Models (edges: 1–3, blocks: 2/5/10, damping: 0.1), and Geometric Random Graphs (radius: 0.08/0.10/0.15)
- **Mechanisms**:
    - Linear additive with homoscedastic noise (weights: 0.25–2.0, 2.0–4.0, and 0.25–4.0; biases: $[-3, 3]$)
    - Linear additive with heteroscedastic noise (same weight and bias ranges, with RFF-based heteroscedastic scale modulation: length scale 10.0, output scale 2.0)
    - RFF additive with RBF kernels (length scale: 5.0–8.0, 8.0–12.0, and 5.0–12.0; output scale: 8.0–15.0, 15.0–22.0, and 8.0–22.0)
    - Heteroscedastic RFF additive with the same kernel ranges and RFF-based heteroscedastic scale modulation (length scale 10.0, output scale 2.0)
- **Noise distributions**: Gaussian, Laplace, Cauchy
- **Interventions**: one intervened variable per interventional row, with signed uniform intervention magnitudes in $[1, 5]$
- **Training dimensions**:
    $d \in \{2, 5, 10, 20, 30, 40, 60, 80, 100\}$

**Strong-prior pretraining.** To test whether broader pretraining distributions improve generalization, we substan-

tially expand the simulator coverage:

- **Graph structures**: Erdős-Rényi (edges per variable: 0.5, 1.0, 2.0, 4.0), Scale-Free and its transpose (edges: 1–2, power: 0.5/1.0/1.5), Stochastic Block Models (edges: 1–2, blocks: 2/5/10, damping: 0.05/0.2), and Geometric Random Graphs (radius: 0.08/0.12/0.16), together with additional structured graph families including staged workflow graphs (stages: 4/5/6, edge probability: 0.12/0.18/0.24, skip probability: 0.02/0.05/0.08, edge cap: 1.1/1.4 per variable), risk-root fork graphs (roots: 1/2/3, branch probability: 0.40/0.55/0.70, local-chain probability: 0.30/0.45/0.60, overlap probability: 0.03/0.08/0.12, edge cap: 1.2/1.6), and an additional sparse scale-free-transpose prior (edges: 1–2, power: 0.8–1.2)
- **Mechanisms**:
    - Linear additive with stratified signal strengths (weights: 0.05–0.5, 0.5–2.0, and 2.0–5.0; biases: $[-2, 2]$; noise scales: 0.05–0.2, 0.2–1.0, and 1.0–2.0)
    - Heteroscedastic linear additive (weights: 0.25–3.0; biases: $[-3, 3]$; RFF-based heteroscedastic scale modulation with length scale 10.0 and output scale 2.0)
    - RFF additive covering both smooth regimes (length scale: 7.0–12.0, output scale: 8.0–20.0) and high-frequency regimes (length scale: 0.5–1.5 and 1.5–3.0, output scale: 0.5–2.0)
    - MLP additive mechanisms in a simple regime (1 hidden layer, hidden dimension: 16/32, activations: Tanh/LeakyReLU, noise scale: 0.1–0.5) and a harder regime (2–3 hidden layers, hidden dimension: 32/64, activation: ReLU, noise scale: 0.2–1.0)
    - Multiplicative linear mechanisms (weights: 0.5–2.0, biases: $[-1, 1]$) with multiplicative noise factors in $[0.8, 1.2]$ or $[0.5, 1.5]$
    - Polynomial additive mechanisms (degree 2 with interactions, weights: 0.1–0.8, biases: $[-0.5, 0.5]$, noise scale: 0.1–0.8)
    - Post-nonlinear models with a linear base (weights: 0.5–2.0, biases: $[-1, 1]$, Gaussian noise scale: 0.2–0.8) followed by Tanh or Sigmoid distortions
    - Additional out-of-distribution neural structural mechanisms with randomized meta-parameters: number of layers sampled from a Gamma-family meta-prior with lower bound 2, hidden dimension sampled from a Gamma-family meta-prior with lower bound 4 and scale up to 100, dropout drawn from a scaled Beta-family meta-prior, structural noise scale drawn from a log-scaled truncated-normal meta-prior with means ranging

from $10^{-4}$ to 0.3, initialization scale drawn from a log-scaled truncated-normal meta-prior with means ranging from $10^{-2}$ to 10.0, number of root causes sampled from a Gamma-family meta-prior with lower bound 2, and activations chosen from {Tanh, Identity, ReLU}; additional binary variations include pre-sampled weights, target-as-effect mode, block-wise dropout, feature sorting, clique-style structure, and random feature rotation

- Additional structured monotone and heavy-tail mechanisms including heavy-tail linear additive (weights: 0.20–2.50, biases: $[-1, 1]$, Student-$t$ noise with df = 4 and Laplace noise, noise scale: 0.10–0.60 and 0.60–1.20), softplus-additive monotone mechanisms (weights: 0.25–2.00, biases: $[-0.8, 0.8]$, noise scale: 0.08–0.50), sigmoid-additive monotone mechanisms (weights: 0.25–1.80, biases: $[-0.8, 0.8]$, noise scale: 0.05–0.35), and threshold-additive mechanisms (weights: 0.25–1.50, biases: $[-0.4, 0.4]$, thresholds: 0.10–0.45, noise scale: 0.05–0.30 and 0.30–0.70)

- **Noise distributions**: Gaussian, Laplace, Uniform, Exponential, Gumbel, Student-$t$ (df = 4), together with multiplicative uniform noise factors for non-additive mechanisms

- **Interventions**: one intervened variable per interventional row, with signed uniform intervention magnitudes typically in $[1, 5]$ for the broad base families, $[1, 4]$ for heavy-tail linear mechanisms, $[1, 3.5]$ for softplus and threshold mechanisms, and $[1, 3]$ for sigmoid, high-frequency RFF, multiplicative, polynomial, and simple/hard MLP mechanisms

- **Training dimensions**: $d \in \{2, 5, 10, 20, 30, 40, 50, 60, 80, 100\}$

The strong-prior configuration expands mechanism diversity and task heterogeneity substantially relative to the weak prior. Crucially, our ablation experiments (Table 6) demonstrate that this richer pretraining distribution yields substantial performance gains, confirming that causal environment diversity is a key driver of generalization in the in-context causal discovery paradigm.

# G. Benchmark Dataset Specifications

We evaluate CausalTab on seven synthetic benchmark families spanning diverse causal mechanisms and graph structures. Unless otherwise specified by the dataset family, graphs use Erdős–Rényi random graphs with average degree following $\bar{d} = 2.0 + 0.05 \cdot (d/100)$, where $d$ is the number of variables. This formula maintains sparsity while adjusting density mildly for higher dimensions.

## G.1. Intervention Method

Each benchmark family is instantiated under **observational-only** and **mixed-interventional** regimes as described in the main experiments. Observational regimes contain no interventional rows ($N_{\text{int}} = 0$). When interventional data are present (mixed regime), we apply *structural* single-variable interventions that propagate to descendants under the structural causal model. Interventions are allocated so that each variable is intervened upon at least once, counts are split as evenly as possible across variables ($\lfloor N_{\text{int}}/d \rfloor$ each with remainder spread across variables), and mixed regimes typically use one intervened variable per interventional sample row. Intervention strengths are drawn from signed uniform ranges (typically $[1.0, 3.0]$ or $[1.0, 5.0]$), representing moderate atomic perturbations.

## G.2. Dataset Specifications

**Linear mechanisms.** Three datasets use linear additive noise models $X_j = \sum_{i \in \text{PA}(j)} w_{ij} X_i + b_j + \varepsilon_j$:

- **linear_gauss**: Edge weights $w_{ij} \sim \text{Uniform}(0.5, 2.0)$, bias $b_j \sim \text{Uniform}(-2, 2)$, Gaussian noise $\varepsilon_j$ with scale $\sim \text{Uniform}(0.5, 1.5)$. Erdős–Rényi graphs.
- **linear_nongauss**: Same weight and bias distributions as above, but noise follows either $\text{Uniform}(-1.5, 1.5)$ or $\text{Exponential}(1.0)$. Erdős–Rényi graphs.
- **linear_graph**: Identical mechanism to linear_gauss, but tests alternative graph topologies: Scale-Free graphs (power = 1.0, average degree = 4.0) and Stochastic Block Models (2/5/10 blocks, edge density 1-3 per variable, damping = 0.1). This tests generalization to hub-and-spoke and modular structures.

**Gaussian Process mechanisms.** Two datasets use nonlinear functions via Random Fourier Features approximating GP priors with RBF kernels. The key distinction lies in the length scale parameter $\ell$:

- **gp_simple**: Length scale $\ell \sim \text{Uniform}(7, 10)$, output scale $c \sim \text{Uniform}(10, 20)$. Large length scales produce smooth, slowly-varying functions. Bias $b \sim \text{Uniform}(-2, 2)$, Gaussian noise with scale $\sim \text{Uniform}(0.5, 1.5)$.
- **gp_hard**: Length scale $\ell \sim \text{Uniform}(0.5, 1.5)$, output scale $c \sim \text{Uniform}(0.5, 2.0)$. Short length scales induce high-frequency oscillations and sharp transitions, significantly increasing approximation difficulty. Bias $b \sim \text{Uniform}(-1, 1)$, Gaussian noise with scale $\sim \text{Uniform}(0.2, 1.0)$.

Both GP datasets use Erdős–Rényi graphs.

**Multiplicative noise.** mul_noise tests robustness to heteroscedasticity via multiplicative noise models: $X_j = \left( \sum_{i \in \mathrm{PA}(j)} w_{ij} X_i + b_j \right) \cdot \varepsilon_j$, where weights $w_{ij} \sim$ Uniform$(0.5, 2.0)$, bias $b_j \sim$ Uniform$(-1, 1)$, and noise multiplier $\varepsilon_j \sim$ Uniform$(0.5, 1.5)$. This violates the additive noise assumption. Erdős–Rényi graphs.

**pfn.** pfn is an additional synthetic causal family included as the seventh benchmark column. Relative to the smoother linear and GP-style families above, it is deliberately sampled from a broader, more heterogeneous mechanism distribution (including highly nonlinear MLP-based structural equations), yielding an out-of-distribution shift at evaluation time. It appears in both observational and mixed-interventional regimes in the synthetic data benchmark (Table 9) and is summarized per $(d, \text{metric})$ in Table 16; the same family is reused for sample-size scaling and high-dimensional scalability experiments in the main text.

**Semantic benchmark inventory and specifications.** The *semantic causal environment benchmark* is a **synthetic**, simulator-grounded suite: each scenario specifies a structural causal model with domain-interpretable variables and directed mechanisms, from which observational and interventional datasets are generated with known adjacencies. It is **not** a field observational "real-world" dataset. Figure 6 and Table 4 summarize empirical scores by domain/regime and by recurring failure-motif tags; the main text reports regime-split aggregates.

*Table 3.* Semantic causal environment benchmark inventory summary. Synthetic, simulator-grounded SCMs with known adjacencies; detailed schema, sampling, generation, validation, and audit specifications appear in the remainder of this appendix.

| Aspect | Summary |
|---|---|
| Suite character | Semantically grounded *synthetic* SCMs simulated from authored JSON; simulator-known DAGs (*not* field observational telemetry). |
| Scale | **10** thematic domains; **100** scenarios (10 per domain); **200** dataset–regime evaluation tasks (obs and int per scenario). |
| Graph size | $\|\mathcal{V}\| \in [8, 17]$. |
| Regimes | obs: observation-only tensors; int: mixed observational and structural-intervention rows with coordinate masks (aligned with Section 3). |
| Sampling scale | Sample sizes drawn uniformly from $\{100, \ldots, 2000\}$; mixed regimes randomize an intervention fraction in $[0.10, 0.30]$ (detailed below). |
| Interventions (mixed) | Balanced variable-level coverage: each variable intervened at least once with near-equal counts (mostly single-target rows). |

## G.3. Package scope and domains

- **Inventory:** 10 thematic domains $\times$ 10 scenarios each $=$ **100** scenario graphs; each scenario yields **two** datasets: an *observation-only* regime and a *mixed* observational–interventional regime (structural interventions), for **200** dataset–regime evaluation instances in total.
- **Domains (readable labels):** healthcare delivery; finance and credit; education; manufacturing; public health; urban transportation; water and sanitation; cybersecurity and IT operations; housing and real estate; government services and policy implementation.

Each authored scenario attaches blueprint metadata, per-variable roles, per-edge mechanism assignments and rationales, difficulty tags, challenge motifs, and semantic intervention handles for interpretation.

## G.4. Graph scale and difficulty

- **Graph size:** every graph has $\|\mathcal{V}\| \in [8, 17]$ nodes; most scenarios are *small-to-medium* graphs rather than very large systems.
- **Per-domain difficulty mix:** within each domain, scenarios follow a fixed composition of **2 easy**, **5 medium**, and **3 hard** graphs; across the full package this yields a *substantial hard subset overall*.

**Semantic rationale.** The benchmark concentrates mass on small-to-medium graphs because difficulty is intended to arise primarily from *semantic bottlenecks*—ambiguous observation channels, delayed or proxy-heavy reporting, screening and detection pathways, queueing and threshold dynamics, and interpretation of interventions—rather than from scaling structural complexity alone. Large graphs are not required for these bottlenecks to be sharp; limiting size keeps scenario authoring and controlled motif repetition tractable while still exercising nonlinear mechanisms and heterogeneous observation families.

## G.5. Regime sampling and intervention construction

**Sample sizes.** Let $d = \|\mathcal{V}\|$ denote the number of variables in the scenario graph.

- **Observation-only regime:** draw $n_{\mathrm{obs}} \sim$ Unif$\{100, \ldots, 2000\}$ and set $n_{\mathrm{int}} = 0$.
- **Mixed regime:** draw total sample size $n \sim$ Unif$\{100, \ldots, 2000\}$ and an intervention ratio $\rho \sim$ Unif$[0.10, 0.30]$. Set $n_{\mathrm{int}} = \max\big(d, \mathrm{round}(n\rho)\big)$ and $n_{\mathrm{obs}} = n - n_{\mathrm{int}}$. If this allocation would violate feasibility (e.g., interventional demand exceeds the sampled total), the generator increases $n$ so that the mixed dataset still contains at least one observational row while respecting the intended intervention coverage.

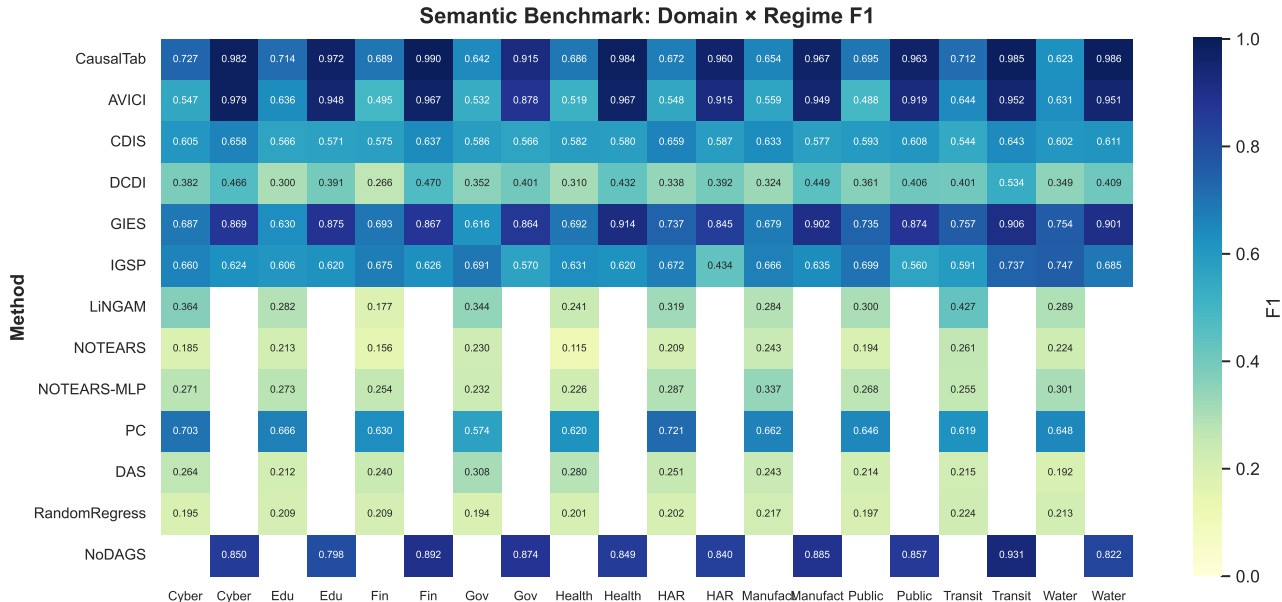

*Figure 6.* **Semantic benchmark domain/regime heatmap.** F1 varies across domains and observational vs. mixed-interventional regimes. Blank cells indicate regimes not applicable or not run for the corresponding method. CausalTab tends to gain most from mixed-interventional evidence, whereas observational panels remain more competitive across baselines; the plot does not imply uniform dominance in every domain.

**Intervention scheduling and values.** Interventions are *structural*: perturbations propagate to descendants in the SCM. When $n_{int} > 0$, generation uses a **balanced full-variable** intervention schedule: every variable is an eligible target; each variable is intervened at least once; per-variable intervention counts differ by at most one; and most interventional rows intervene on a *single* variable. **Named intervention handles** in the scenario narrative are *reader-facing semantic anchors*; they are not the only variables intervened in the dataset, because balanced coverage across all variables avoids degenerate "single-handle-only" mixed regimes.

Intervention strengths use **strong two-sided shifts**: values are centered near $\pm 1.75$ with Gaussian perturbations at spread roughly 0.30. **Semantic rationale.** Balanced coverage prevents mixed data from becoming trivially sparse or dominated by one narrated lever while still preserving interpretable intervention semantics for readers. Interventions are informative but *not uniformly revealing*: noisy, censored, proxy, or delayed observation channels can still obscure post-interventional patterns, mirroring realistic semantic friction.

### G.6. Mechanism vocabulary (response-shape templates)

Mechanism labels describe **canonical response-shape templates** applied to a *randomized* aggregated parent sig-

nal at each node: graph-specific signed weights on incoming edges, node offsets/biases, structural noise draws, and observation rendering noise vary across edges and scenarios. Two nodes sharing a label therefore need not behave identically numerically; the label specifies the qualitative nonlinearity imposed on the latent pre-activation.

**Primary shape families (templates).**

- **linear:** identity response on weighted parent aggregation.
- **softplus:** monotone softplus-type activation, $\approx \log 1 p(\exp(\text{clip}(x, -8, 8))) - \log 2$.
- **sigmoid:** bounded monotone response, $\approx 2/(1+e^{-x}) - 1$.
- **threshold:** positive-part thresholding, $\approx \max(x - 0.25, 0)$.
- **saturation:** saturating response, $\approx \tanh(x/1.5)$.
- **step:** binary thresholding around 0.
- **reciprocal_like:** smooth sign-preserving compression, $\approx \tanh(x)$.
- **u_shape:** U-shaped response, $\approx x^2 - 1$.
- **heavy_tail_driver:** heavy-tail-sensitive mapping, $\approx \text{sign}(x)\sqrt{|x| + 10^{-6}}$.

**Rare optional shapes.** `piecewise`, `interaction`, and `accumulation` appear only as negligible exceptions in the current package.

**Package-level edge-mechanism proportions (constraints).**

- **linear:** about 50%–60% of edges.
- **softplus + saturation:** together about 30%–40%.
- **sigmoid:** about 5%–10%.
- **threshold + step:** together about 5%–10%.
- **reciprocal_like + u_shape + heavy_tail_driver:** together at most $\approx 5\%$.
- **optional rare mechanisms:** together at most $\approx 5\%$.

**Semantic rationale.** Linear mechanisms dominate because many domain narratives admit approximately local, monotone causal influence over the modeled operating range. Softplus and saturation capture activation and ceiling effects common in queues, throughput limits, adherence, and screening uptake. Sigmoid-like shapes suit bounded middle-range outcomes (rates, probabilities, completion fractions). Threshold and step shapes are used sparingly for gating, escalation, screening cutoffs, and policy triggers. Reciprocal-like, U-shaped, and heavy-tail-driver templates are intentionally rare: they encode specific semantic motifs rather than blanket exotic nonlinearity.

### G.7. Latent structural noise families

Latent structural disturbances are drawn from a small vocabulary (applied with randomized scales tied to each scenario):

- **gaussian:** zero-mean Gaussian structural noise.
- **heteroscedastic:** Gaussian noise multiplied by a random scale factor $\approx 0.7 + U(0, 1)$.
- **heavy_tail:** Student-$t$–style noise with degrees of freedom 3, scaled by $\approx 0.6$.
- **asymmetric:** Gaussian component plus a one-sided exponential component.
- **mixture:** Gaussian baseline with an additional "burst" component activated with probability $\approx 0.14$ at scale $\approx 2\times$ the baseline.

**Typical scales:** structural noise scales for latent equations concentrate roughly in $[0.14, 0.32]$; the package is balanced so no single noise family dominates every scenario.

**Semantic rationale.** Gaussian noise represents routine operational variability; heteroscedastic noise captures load-dependent volatility; asymmetric components model delays and one-sided disruptions; mixture noise injects occasional bursts (failures, surges); heavy tails support rare high-impact instability without pathologizing the entire benchmark.

### G.8. Observation channels and corruption families

Observation families specify how latent variables are rendered into tabular columns (with scenario-specific noise scales).

**Main observation families.** `identity`, `bounded`, `positive`, `ordinal`, `contaminated_identity`, `missing_to_zero_identity`, `missing_to_zero_positive`, `zero_inflated_positive`, `censored_positive`.

**Rare optional observation families.** `count_like`, `rate_like`, `proxy_noisy`, `missingness_corrupted` (negligible prevalence overall).

**Package-level observation proportions (constraints).**

- **bounded:** about 45%–55% of variable observations.
- **positive:** about 30%–40%.
- **identity:** about 5%–15%.
- **ordinal:** at most $\approx 5\%$.
- **all "hard" observation families combined:** about 2%–10%.
- A minority of graphs include at least one hard family; only a small subset include multiple hard families simultaneously; the majority stay "clean" in the sense of using primarily identity/bounded/positive/ordinal channels.

**Rendering behavior (qualitative).**

- **identity:** additive Gaussian observation noise on centered latent channels.
- **bounded:** logistic-style squashing toward $[-1, 1]$.
- **positive:** softplus-style strictly positive rendering.
- **ordinal:** thresholded discretization with a small fixed cut set.
- **censored_positive:** positive rendering with upper censoring.
- **zero_inflated_positive:** positive rendering with $\approx 10\%$ zero inflation.
- **contaminated_identity:** identity rendering with $\approx 5\%$ contamination spikes.
- **missing_to_zero_identity / missing_to_zero_positive:** $\approx 10\%$ missingness mapped to zero.

**Observation noise scales** (post-rendering) typically fall roughly in $[0.04, 0.159]$ with median near $0.076$.

**Semantic rationale.** Bounded channels suit normalized scores, compliance fractions, readiness indices, and probability-like summaries; positive channels suit volumes, queues, delays, concentrations, and costs; identity suits centered latent process drivers; ordinal suits coarse grading tiers. Hard corruptions model measurement error, alarms with rare gross errors, screening and follow-up pipelines where absence is confounded with zeros, and top-coded reporting. These harder channels are *concentrated* on mea-

surement, detection, alarm, screening, proxy, and follow-up style nodes rather than applied uniformly to every variable, so ambiguity stays tied to semantically plausible friction points.

### G.9. Edge weights and scenario-specific authoring

The semantic package is **not** a single anonymous random-graph template with arbitrary weights: edges are authored with graph-specific semantic roles. **Design targets** absolute structural weights roughly in $[0.20, 0.85]$, with central process drivers typically stronger than reporting, proxy, or bookkeeping channels. **Realized package statistics** for absolute weights span roughly $[0.30, 1.18]$ with median near 0.718—still consistent with hand-authored heterogeneity rather than a degenerate constant template.

**Semantic rationale.** Heterogeneous strengths preserve scenario-specific bottlenecks (e.g., weak proxy edges vs. strong operational drivers) and keep identification challenges tied to narrative structure rather than to a uniform synthetic weight prior.

### G.10. Recurring blueprint motifs and analysis-slice tags

**Blueprint-level motif families (examples).** Recurring cross-domain narrative templates include: triage–threshold–queue; forecast–buffer–imbalance; calibration–drift–alarm; screening–detection–followup; contamination–spread–control; engagement–fatigue–mastery; pricing–friction–conversion; throughput–reciprocal–process.

**Analysis-slice vocabulary (failure-mode tags).** Evaluation uses standardized slice labels such as:

- `legacy_triage_threshold`,
- `legacy_measurement_failure`,
- `legacy_detection_failure`,
- `legacy_followup_breakdown`,
- `legacy_alarm_drift`,
- `legacy_proxy_interpretation`,
- `legacy_queue_pressure`,
- `legacy_reciprocal_process`,
- `legacy_heavy_tail_instability`, and
- `legacy_mixed_regime_ambiguity`.

These tags denote **recurring semantic bottlenecks**, not geographic domains: graphs may carry multiple slices, and slices recur across domains to align surface narratives with a controlled library of challenge types.

**Why this matters:** motif families and slice tags create reusable stress patterns (threshold congestion, missed detection, broken follow-up, drifted alarms, proxy ambiguity, queue overload, reciprocal feedback, heavy-tail instability,

observational–interventional ambiguity) while preserving interpretable variable meanings.

### G.11. Checklist-guided packaging criteria

A packaging checklist enforces coverage (10/10/100), regime sampling consistency, graph-size limits (8–17 nodes), schema completeness for challenge profiles and analysis slices, semantic authoring rules (non-boilerplate rationales, meaningful intervention handles), blueprint fidelity, mechanism and observation mix targets, weight diversity, and numerical rendering checks. These criteria complement—but do not replace—the structured LLM-aided consistency and documentation review in Section G.13.

### G.12. Semantic benchmark results: failure motifs

Table 4 lists mean F1 by analysis-slice tags associated with recurring semantic bottlenecks.

These motif tags are not disjoint datasets, but recurring semantic failure-mode annotations (graphs can carry multiple slice labels). Across the mixed-interventional slices, CausalTab achieves consistently high F1 across motifs, indicating that its gains are not concentrated in a single semantic pattern but persist across heterogeneous intervention and mechanism settings. The relatively narrow performance range across motifs further suggests stable behavior under different semantic bottlenecks. Observation-only slices are naturally more competitive due to directional ambiguity, but CausalTab still maintains stable performance across motifs, supporting the robustness of pretrained dataset-to-graph inference in semantically structured environments.

### G.13. Structured LLM-aided consistency audit

Packaging many heterogeneous authored scenarios raises auxiliary risks—schema drift, templated prose fields, or inconsistent textual hardness tags—that can distract readers even when numerical SCM instantiation stays faithful. We therefore run a structured LLM-aided audit that operates **independently** of CausalTab scores or any other method rankings, using **Claude** as the review model.

The audit is organized as a documentation and consistency check rather than exhaustive human semantic certification. Concretely, it covers: **(i)** protocol fidelity checks against the packaged authoring specification; **(ii)** generator alignment with documented sampling and intervention rules; **(iii)** aggregate distributional consistency checks against benchmark-level summaries; and **(iv)** sampled scenario-level semantic coherence of inspected JSON relative to blueprint metadata, stratified across domains and difficulty tiers. The review ingests the benchmark protocol, gener-

*Table 4.* **Semantic benchmark breakdown by failure motif.** Mean F1 grouped by recurring semantic bottlenecks or failure motifs, reported separately for observation-only and mixed-interventional regimes.

**Observation-only**

| Motif | CausalTab | AVICI | CDIS | DCDI | GIES | IGSP | LiNGAM | NOTEARS | NOTEARS-MLP | PC | DAS | RandomRegress |
|---|---|---|---|---|---|---|---|---|---|---|---|---|
| alarm_drift | 0.640 | 0.574 | 0.737 | 0.377 | 0.789 | 0.705 | 0.246 | 0.199 | 0.315 | 0.627 | 0.257 | 0.188 |
| detection_failure | 0.674 | 0.537 | 0.586 | 0.328 | 0.714 | 0.679 | 0.310 | 0.219 | 0.287 | 0.665 | 0.251 | 0.207 |
| followup_breakdown | 0.675 | 0.553 | 0.566 | 0.343 | 0.704 | 0.702 | 0.303 | 0.204 | 0.271 | 0.670 | 0.243 | 0.212 |
| heavy_tail_instability | 0.682 | 0.548 | 0.466 | 0.332 | 0.570 | 0.527 | 0.301 | 0.142 | 0.229 | 0.593 | 0.217 | 0.210 |
| measurement_failure | 0.691 | 0.560 | 0.608 | 0.330 | 0.720 | 0.676 | 0.334 | 0.229 | 0.288 | 0.665 | 0.247 | 0.208 |
| mixed_regime_ambiguity | 0.681 | 0.560 | 0.595 | 0.338 | 0.698 | 0.664 | 0.303 | 0.203 | 0.270 | 0.649 | 0.242 | 0.206 |
| proxy_interpretation | 0.688 | 0.566 | 0.589 | 0.336 | 0.706 | 0.681 | 0.302 | 0.201 | 0.249 | 0.652 | 0.236 | 0.208 |
| queue_pressure | 0.650 | 0.548 | 0.606 | 0.354 | 0.647 | 0.624 | 0.257 | 0.145 | 0.238 | 0.608 | 0.230 | 0.203 |
| reciprocal_process | 0.682 | 0.548 | 0.466 | 0.332 | 0.570 | 0.527 | 0.301 | 0.142 | 0.229 | 0.593 | 0.217 | 0.210 |
| triage_threshold | 0.701 | 0.594 | 0.611 | 0.338 | 0.706 | 0.659 | 0.237 | 0.198 | 0.243 | 0.630 | 0.242 | 0.192 |

**Mixed-interventional**

| Motif | CausalTab | AVICI | CDIS | DCDI | GIES | IGSP | NoDAGS |
|---|---|---|---|---|---|---|---|
| alarm_drift | 0.974 | 0.923 | 0.738 | 0.442 | 0.926 | 0.656 | 0.915 |
| detection_failure | 0.972 | 0.943 | 0.600 | 0.409 | 0.907 | 0.564 | 0.859 |
| followup_breakdown | 0.978 | 0.950 | 0.591 | 0.440 | 0.886 | 0.637 | 0.849 |
| heavy_tail_instability | 0.935 | 0.932 | 0.460 | 0.403 | 0.839 | 0.581 | 0.849 |
| measurement_failure | 0.973 | 0.946 | 0.623 | 0.418 | 0.900 | 0.584 | 0.877 |
| mixed_regime_ambiguity | 0.971 | 0.943 | 0.604 | 0.435 | 0.882 | 0.611 | 0.864 |
| proxy_interpretation | 0.976 | 0.948 | 0.608 | 0.445 | 0.880 | 0.613 | 0.853 |
| queue_pressure | 0.969 | 0.935 | 0.585 | 0.477 | 0.851 | 0.645 | 0.841 |
| reciprocal_process | 0.935 | 0.932 | 0.460 | 0.403 | 0.839 | 0.581 | 0.849 |
| triage_threshold | 0.971 | 0.936 | 0.616 | 0.466 | 0.842 | 0.647 | 0.858 |

ator logic, packaged metadata, benchmark-level summary statistics, and a stratified set of scenario JSON files.

Overall, the findings support treating the suite as a semantically grounded synthetic benchmark whose generator behavior is broadly consistent with the documented protocol and whose inspected scenarios remain structurally coherent relative to their blueprint metadata, subject to the usual limitations of automated review. The audit also surfaced minor metadata-quality issues (e.g., templated free-text fields or occasional inconsistent auxiliary descriptors); these primarily affect narrative annotations rather than the underlying SCM graphs, sampling code, or evaluation pipeline.

LLM-assisted auditing should be interpreted as scalable **documentation and consistency support**—not exhaustive semantic certification, not a substitute for domain experts, and not evidence of field observational realism or operational validity—but useful structured hygiene alongside checklist-driven packaging. Relative method rankings on the benchmark may additionally depend on thresholding choices and runtime budgets (Section C).

### G.14. Illustrative domain-grounded scenarios

We highlight two authored scenarios to show how semantics attach to SCM structure without reproducing full specifications.

**Healthcare delivery (screening–detection–follow-up blueprint).** A diabetes follow-up coordination scenario

instantiates a screening–detection–follow-up motif on a nine-variable graph. Nodes carry interpretable roles such as latent demand, access barriers, screening signals, detection flags, follow-up capacity, and completed workups. Named intervention handles (*workup action*, *follow-up slot capacity*) aid interpretation while mixed-regime data still use balanced interventions across all variables for structural identifiability. Challenge slices emphasize detection failures, follow-up breakdowns, measurement ambiguity, proxy interpretation, and mixed-regime ambiguity—canonical bottleneck templates reused across domains.

**Urban transportation (triage–threshold–queue blueprint).** A signal-timing scenario authors operations as a triage–threshold–queue motif with twelve variables spanning demand load, intake signals, queue congestion, priority scoring, threshold decisions, service starts, backlog spillover, and downstream delay harms. Named intervention handles (*priority score*, *service start*) align with operational levers operators discuss, while challenge slices highlight threshold pressures, queue congestion, follow-up breakdowns, proxy ambiguity, and mixed-regime tension—reflecting recurring workload-motif stressors.

*Table 5.* Illustrative semantic scenarios (selected summary fields).

| Scenario (abbrev.) | Domain | Blueprint | $|\mathcal{V}|$ |
|---|---|---|---|
| Diabetes follow-up coordination | Healthcare delivery | Screening–detection–follow-up | 9 |
| Signal-timing intervention | Urban transportation | Triage–threshold–queue | 12 |

# H. Additional Experimental Results

This appendix provides comprehensive experimental details and extended results that supplement the main paper.

## H.1. Prior strength ablation

We conduct an ablation study isolating pretraining prior strength.

**Prior strength.** We compare two variants that differ only in causal environment coverage during pretraining. The **weak-prior** variant focuses on linear additive and GP/RFF additive mechanisms (including heteroscedastic noise) over multiple graph distributions (e.g., ER, SF, SBM, GRG). The **strong-prior** variant substantially expands both (i) graph-structure coverage (wider sparsity/density and broader structural parameters) and (ii) mechanism/noise diversity (e.g., MLP mechanisms, multiplicative noise, polynomial interactions, post-nonlinear models, and richer non-Gaussian/asymmetric noises).

**Results.** Table 6 summarizes the findings. The strong-prior full model substantially improves over the weak-prior variant across SHD, SID, and F1, demonstrating that broader causal environment coverage during pretraining directly translates to stronger out-of-distribution generalization. This confirms that rich pretraining task diversity is crucial for the CDFM paradigm.

*Table 6.* **Prior-strength ablation.** Effects of weak vs. strong pretraining priors under the same evaluation protocol.

| Variant | SHD↓ | SID↓ | F1↑ |
|---|---|---|---|
| Weak prior | 26.0 | 40.4 | 0.659 |
| Strong prior (full) | 3.6 | 6.6 | 0.888 |

## H.2. Predicting Graph Statistics from Embeddings

To quantify how much structural information is captured by embeddings, we first examine embedding distances between different causal substructures, then test whether pooled embeddings can predict global graph statistics. Figure 7 shows average embedding distances for different causal relationships. Distances increase monotonically with causal path length (1-hop: 17.2 < 2-hop: 20.1 < 3+hop: 26.3), while structures with shared parents or children (forks: 14.6, colliders: 15.8) maintain the smallest distances, and independent node pairs (24.8) are well-separated. This demonstrates that the embedding space preserves causal proximity: nodes closer in the causal graph have more similar representations.

We further test whether pooled embeddings can predict global graph properties. We train lightweight Ridge re-

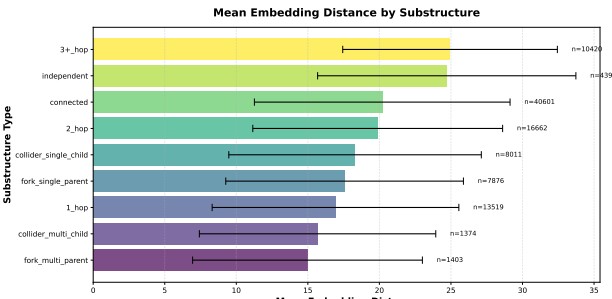

*Figure 7.* **Embedding distances vs. causal substructures.** Distances increase monotonically with path length, structures with shared parents/children have smallest distances.

*Table 7.* Graph statistic prediction performance. Embedding probes reduce MAE by 30-62% compared to raw features.

| Target | Embed MAE↓ | Raw MAE | Embed $R^2$↑ | MAE Gain |
|---|---|---|---|---|
| Edge count $|E^\star|$ | 7.82 | 12.67 | 0.865 | $-38.3\%$ |
| Average degree $\bar{d}$ | 0.42 | 1.09 | 0.909 | $-61.7\%$ |
| Max in-degree | 0.96 | 1.79 | 0.814 | $-46.7\%$ |
| DAG depth | 0.98 | 1.41 | 0.716 | $-30.3\%$ |

gression (Hoerl & Kennard, 2000) probes to predict four statistics from pooled embeddings $\widetilde{\mathbf{h}}_{\text{set}} = \text{mean}(\{\widetilde{\mathbf{h}}_i\}_{i=1}^d)$: edge count $|E^\star|$, average degree $\bar{d}$, maximum in-degree, and DAG depth. We compare against raw data features (correlation matrix statistics, means, standard deviations) and random projections as baselines.

Figure 8 shows that embedding-based probes achieve strong prediction accuracy across all targets ($R^2 > 0.72$). Table 7 quantifies the performance: embeddings reduce MAE by 30-62% compared to raw feature baselines, with the largest gain for average degree (61.7% reduction). Notably, direction-sensitive statistics (maximum in-degree, DAG depth) are also accurately predicted, indicating embeddings encode directed topological structures beyond undirected connectivity patterns. These results demonstrate that CausalTab's learned representations contain reusable structural information supporting downstream tasks.

## H.3. Computational Efficiency

We report mean wall-clock time per sample on the synthetic benchmark, aggregated across graph sizes $d \in \{5, 10, 20\}$ and the seven data families used in Section 4. Figure 9 visualizes these per-method means in seconds; entries shown as `>120s` denote averages above a 120 s plotting cap rather than exact unattainable values on the axis. CausalTab performs amortized inference in one forward pass, whereas many classical and neural baselines require per-instance conditional testing, combinatorial or continuous search, or iterative fitting, which yields substantially higher average

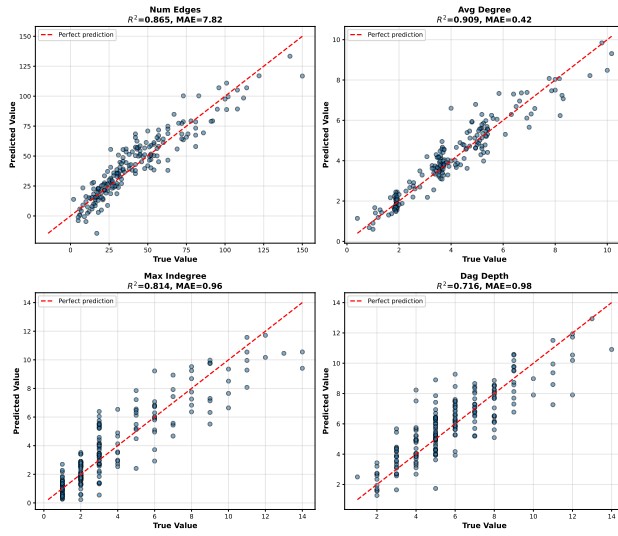

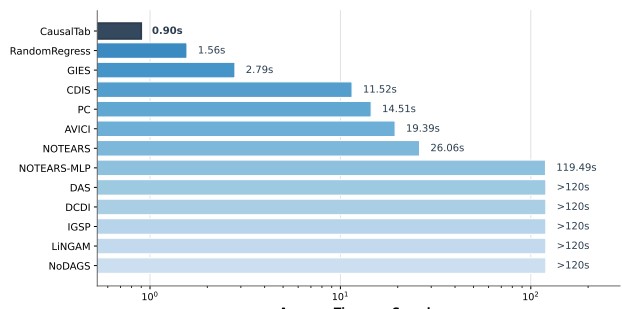

Figure 9. **Computational efficiency.** Mean wall-clock *time per sample* (seconds) on the synthetic benchmark for each listed method, aggregated over the same graph sizes $d \in \{5, 10, 20\}$ and seven data families as the main synthetic evaluation (Section 4). Methods whose mean exceeds a 120 s display threshold are shown as `>120s`. CausalTab performs amortized inference in a single forward pass, whereas several baselines rely on per-instance testing, search, or iterative optimization and exhibit much larger per-sample means in this view. Reported seconds are produced by our benchmarking harness and can vary with hardware and implementation choices.

Figure 8. **Predicting graph statistics from embeddings.** True vs. predicted values for four graph properties (edge count, average degree, max in-degree, and DAG depth). Simple probes on our learned embeddings achieve strong fits ($R^2 \geq 0.71$), substantially outperforming raw-feature baselines.

time per sample under the same measurement protocol. The spread in the figure highlights a stable qualitative gap between amortized prediction and several heavier baselines, without relying on a particular numeric speedup factor. Absolute timings and relative ordering can shift with hardware, batching, solver settings, and engineering optimizations, but the contrast between single-pass prediction and per-instance structure search remains evident in this aggregate view.

### H.4. Embedding Quality Analysis

To understand when CausalTab succeeds or fails, we analyze embedding quality across three difficulty factors: sample size ($N \in \{10, 100, 1000\}$), dimension ($d \in \{5, 10, 30, 100\}$), and graph complexity (edge density $p \in \{0.05, 0.1, 0.2, 0.4\}$). For each condition, we generate 50 random DAGs and evaluate three metrics: (1) **Separability**: mean distance between non-edges minus mean distance between true edges, (2) **k-NN Accuracy**: proportion of k-nearest neighbors that are true graph neighbors, (3) **Effective Dimensionality**: number of principal components explaining 90% variance.

Table 8 shows the results across all conditions. **Dimension is the primary bottleneck.** From $d = 5$ to $d = 100$, k-NN accuracy drops from 1.000 to 0.383 ($-62\%$) and separability from 7.44 to 1.20 ($-84\%$), while effective dimensionality inflates from 3 to 23 ($+667\%$). Performance degrades sharply between $d = 10$ (k-NN = 0.853) and $d = 30$ (k-NN = 0.489), suggesting a critical threshold around $d \approx 15$–20.

**Sample size shows diminishing returns beyond $N = 100$.** Increasing $N$ from 10 to 100 yields dramatic improvements: separability increases by 58% and k-NN accuracy by 35%. However, further increasing to $N = 1000$ brings marginal gains ($< 5\%$), suggesting an effective sample size of roughly $N \approx 5d$ suffices—a sample efficiency advantage over classical methods requiring $N \gg d^2$.

**Graph complexity causes linear degradation.** As edge density increases from $p = 0.05$ to $p = 0.4$, separability decreases from 7.33 to 1.76 ($-76\%$) and k-NN accuracy from 0.923 to 0.597 ($-35\%$). Unlike dimension (exponential collapse), graph complexity exhibits predictable linear degradation.

These results suggest CausalTab performs reliably when $d \lesssim 20$, $N \gtrsim 5d$, and graphs are moderately sparse ($p \lesssim 0.2$). The sharp degradation beyond $d \approx 20$ indicates that scaling to high-dimensional settings may require architectural innovations rather than simply more training data.

## I. Hyperparameter Configuration

Unless noted otherwise, the methods below are evaluated using the benchmark wrapper's current direct/default configuration rather than a tune-then-test validation sweep. The appendix therefore reports the fixed wrapper settings used in the current benchmark pipeline. Continuous-output methods are binarized using each wrapper's default decoding rule consistent with the evaluation pipeline.

*Table 8.* Embedding quality metrics across difficulty levels. **Sep**: Separability. **k-NN**: k-Nearest neighbor accuracy. **Eff.Dim**: Effective dimensionality.

| Factor | Level | Sep↑ | k-NN↑ | Eff.Dim↓ |
|--------|-------|------|-------|----------|
| Sample Size | $N = 10$ | 2.02 | 0.441 | 9 |
| | $N = 100$ | 3.19 | 0.594 | 12 |
| | $N = 1000$ (r1) | 3.63 | 0.596 | 11 |
| | $N = 1000$ (r2) | 3.22 | 0.599 | 11 |
| Dimension | $d = 5$ | 7.44 | 1.000 | 3 |
| | $d = 10$ | 4.94 | 0.853 | 5 |
| | $d = 30$ | 2.73 | 0.489 | 15 |
| | $d = 100$ | 1.20 | 0.383 | 23 |
| Graph Complexity | Very sparse ($p = 0.05$) | 7.33 | 0.923 | 7 |
| | Sparse ($p = 0.1$) | 5.07 | 0.765 | 9 |
| | Medium ($p = 0.2$) | 3.50 | 0.618 | 11 |
| | Dense ($p = 0.4$) | 1.76 | 0.597 | 11 |

## I.1. Method-Specific Hyperparameters

**CausalTab.** Pretrained transformer-based amortized dataset-to-graph predictor; reported numbers use the final pretrained checkpoint from this paper, without per-instance retraining or test-time optimization.

**AVICI.** Official pretrained weights released by the authors; no per-instance optimization.

**PC.**

- Significance level $\alpha = 0.05$
- Conditional independence testing: Fisher-$Z$ / Gaussian tests
- Orientation: liberal strategy (the wrapper supports other choices; reported benchmarks use these defaults)

**GIES.**

- Penalty: BIC-style default ($\lambda = -1.0$ in the wrapper)
- Orientation: liberal strategy

**IGSP.**

- Gaussian conditional independence testing
- Observational significance level $\alpha = 10^{-3}$
- Invariance-test significance $\alpha_{\text{inv}} = 10^{-3}$

**LiNGAM.** DirectLiNGAM through the current wrapper with standard decoding/default handling.

**CDIS.** Library default discovery procedure with liberal orientation in the current direct wrapper.

**NOTEARS.** Linear NOTEARS with $\ell_1$ coefficient 0.1.

**NOTEARS-MLP.** Hidden width 10, $\lambda_1 = 0.01$, $\lambda_2 = 0.01$.

**NoDAGS.**

- Acyclicity coefficient: $10^{-3}$
- Learning rate: $10^{-2}$
- Depth: 0 hidden layers
- Training: 200 epochs, batch size 1024
- Minimum interventional samples per regime: 10

**DCDI.**

- Encoder: 2 layers, hidden width 16
- Flow: 2 layers, hidden width 16
- Activation: Leaky ReLU
- Optimizer: RMSprop, learning rate $10^{-3}$
- Regularization coefficient: 0.1
- Inputs normalized in the wrapper
- Train/test split: 80/20 within each sampled dataset
- Mini-batch size: $\min(64, N_{\text{train}})$
- Augmented Lagrangian optimization with small initial multipliers and patience-based early stopping
- Maximum optimizer iterations: 30,000

**DAS.**

- $\eta_G = 10^{-3}$, $\eta_H = 10^{-3}$, $\alpha = 0.05$
- Pruning enabled
- Splines: degree 3 with 10 basis functions
- Modest parent-set constraints per wrapper defaults

**RandomRegress.** Random-order regression sanity baseline; deliberately weak structural control.

# J. Impact Statement

This paper presents work whose goal is to advance the field of Machine Learning. There are many potential societal consequences of our work, none which we feel must be specifically highlighted here.

# K. Full Synthetic Benchmark Results

### K.1. Macro-averaged benchmark summary

### K.2. Performance Heatmap Across All Benchmarks

Figure 10 heatmaps F1 for each method on each synthetic dataset-regime column included in the figure.

Note that some baselines are not applicable to both regimes: NOTEARS, NOTEARS-MLP, LiNGAM, PC, DAS, and RandomRegress are evaluated only in the observational setting, while NoDAGS is evaluated only in the mixed-interventional setting.

*Table 9.* **Synthetic data benchmark.** Overall performance macro-averaged over all (dataset family $\times$ $d$) settings with equal weight. **Obs** uses 1000 observational samples; **Mixed** uses 800 observational + 200 interventional samples. Some methods are not applicable to both regimes: NOTEARS, NOTEARS-MLP, LiNGAM, PC, DAS, and RandomRegress are evaluated only on observational data, while NoDAGS is evaluated only on mixed-interventional data. **Rank** is the average rank across all configurations (lower is better). Detailed per-$(d, \text{family})$ scores appear in Section K.3.

| Method | Observation | | | | Mixed-intervention | | | |
| | Rank↓ | F1↑ | SHD↓ | SID↓ | Rank↓ | F1↑ | SHD↓ | SID↓ |
| --- | --- | --- | --- | --- | --- | --- | --- | --- |
| RandomRegress | 11.6 | 0.240 | 63.3 | 76.1 | – | – | – | – |
| DAS | 3.4 | 0.699 | 10.7 | 20.6 | – | – | – | – |
| LiNGAM | 8.9 | 0.420 | 15.7 | 30.4 | – | – | – | – |
| PC | 6.9 | 0.496 | 14.1 | 39.6 | – | – | – | – |
| CDIS | 8.9 | 0.429 | 15.0 | 41.6 | 5.7 | 0.500 | 14.0 | 38.7 |
| GIES | 5.0 | 0.599 | 11.6 | 28.2 | 2.6 | 0.780 | 8.1 | 16.8 |
| IGSP | 5.6 | 0.554 | 15.5 | 41.8 | 5.4 | 0.540 | 13.6 | 37.0 |
| NOTEARS | 6.1 | 0.560 | 9.3 | 17.8 | – | – | – | – |
| NOTEARS-MLP | 5.6 | 0.585 | 12.0 | 24.7 | – | – | – | – |
| NoDAGS | – | – | – | – | 4.7 | 0.573 | 12.7 | 27.0 |
| DCDI | 8.3 | 0.430 | 14.5 | 31.1 | 5.9 | 0.459 | 12.5 | 26.4 |
| AVICI | 6.0 | 0.551 | 11.6 | 24.8 | 2.7 | 0.844 | 6.3 | 9.4 |
| CausalTab | 1.9 | 0.824 | 5.1 | 11.1 | 1.0 | 0.952 | 2.1 | 2.0 |

## K.3. Detailed Per-Dataset Results

This section provides complete numerical results for all benchmarks across different dimensions and evaluation metrics. Each table reports F1 score, Structural Hamming Distance (SHD), and Structural Intervention Distance (SID) for both observational and mixed-interventional settings.

Each table is organized as follows:

- Columns are grouped by evaluation setting (Observation vs. Mixed-intervention)
- Within each setting, results are shown for three dimensions ($d \in \{5, 10, 20\}$)
- For each dimension, we report three metrics: F1 (higher is better), SHD (lower is better), and SID (lower is better)
- "–" indicates the method is not applicable to that setting

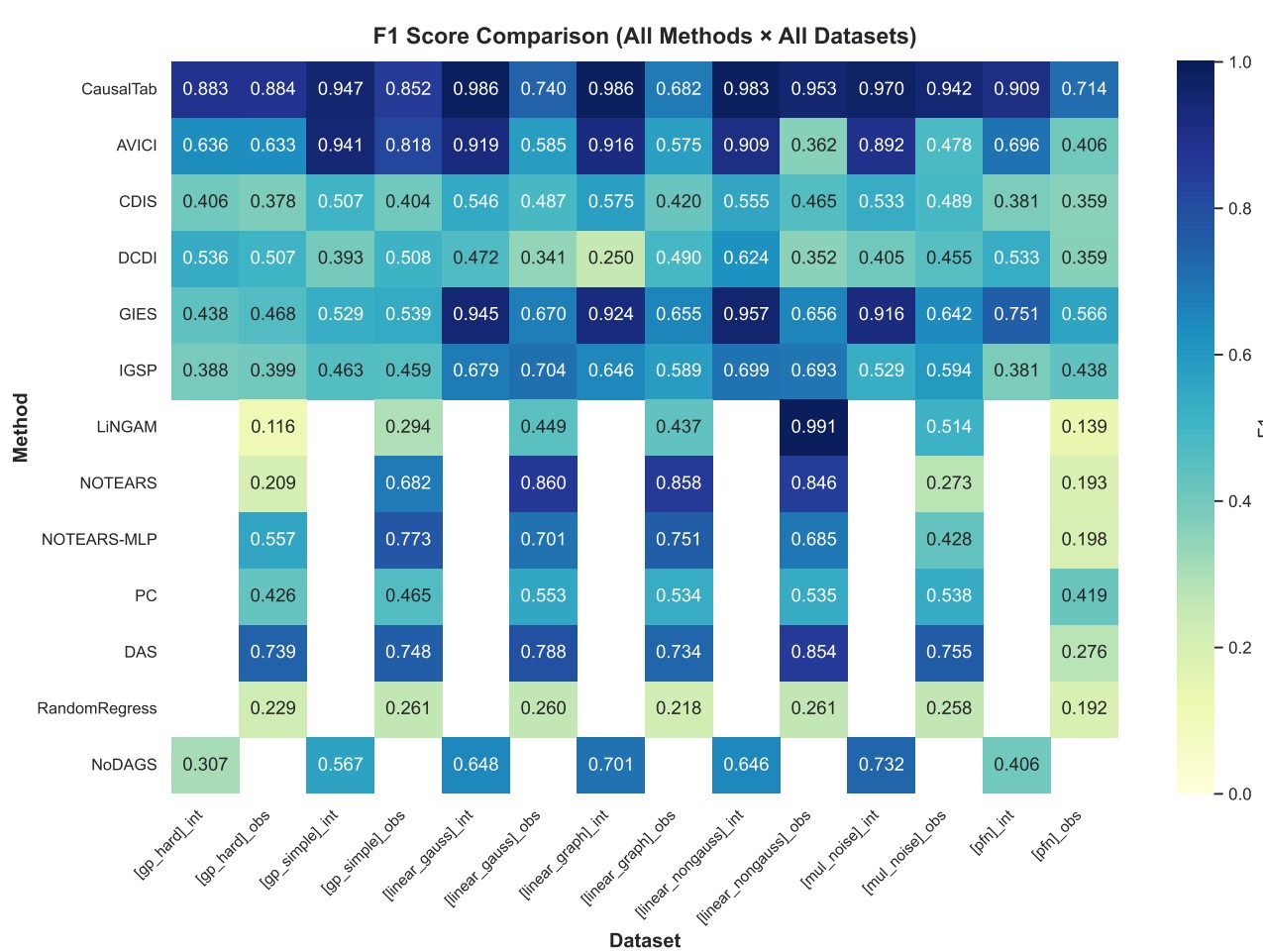

*Figure 10.* **Detailed F1 heatmap across dataset-regime settings.** Each cell reports F1 for a method on one observational or mixed-interventional dataset setting; blank cells denote methods not applicable or not run under that regime.

*Table 10.* Detailed results for `gp_hard` across different graph sizes ($d$) and metrics.

| Method | Observation | | | | | | | | | Mixed-intervention | | | | | | | | |
|---|---|---|---|---|---|---|---|---|---|---|---|---|---|---|---|---|---|---|
| | $d=5$ | | | $d=10$ | | | $d=20$ | | | $d=5$ | | | $d=10$ | | | $d=20$ | | |
| | F1 | SHD | SID | F1 | SHD | SID | F1 | SHD | SID | F1 | SHD | SID | F1 | SHD | SID | F1 | SHD | SID |
| RandomRegress | 0.30 | 7 | 9 | 0.23 | 33 | 41 | 0.16 | 140 | 169 | – | – | – | – | – | – | – | – | – |
| DAS | 0.83 | 1 | 2 | 0.71 | 8 | 13 | 0.61 | 25 | 61 | – | – | – | – | – | – | – | – | – |
| LiNGAM | 0.11 | 5 | 6 | 0.13 | 14 | 24 | 0.11 | 37 | 84 | – | – | – | – | – | – | – | – | – |
| PC | 0.47 | 3 | 6 | 0.43 | 12 | 26 | 0.38 | 34 | 99 | – | – | – | – | – | – | – | – | – |
| CDIS | 0.38 | 4 | 6 | 0.40 | 13 | 28 | 0.36 | 35 | 108 | 0.45 | 4 | 6 | 0.42 | 12 | 25 | 0.35 | 36 | 106 |
| GIES | 0.49 | 3 | 6 | 0.46 | 13 | 28 | 0.44 | 34 | 108 | 0.48 | 4 | 6 | 0.43 | 13 | 30 | 0.38 | 40 | 121 |
| IGSP | 0.44 | 3 | 6 | 0.39 | 13 | 28 | 0.37 | 34 | 101 | 0.43 | 3 | 6 | 0.40 | 11 | 27 | 0.34 | 36 | 102 |
| NOTEARS | 0.25 | 4 | 5 | 0.21 | 13 | 23 | 0.16 | 35 | 81 | – | – | – | – | – | – | – | – | – |
| NOTEARS-MLP | 0.56 | 3 | 4 | 0.57 | 9 | 16 | 0.53 | 25 | 65 | – | – | – | – | – | – | – | – | – |
| NoDAGS | – | – | – | – | – | – | – | – | – | 0.38 | 4 | 5 | 0.31 | 13 | 21 | 0.20 | 38 | 89 |
| DCDI | 0.33 | 4 | 3 | 0.68 | 13 | 50 | – | – | – | 0.55 | 4 | 7 | 0.53 | 17 | 26 | – | – | – |
| AVICI | 0.72 | 2 | 3 | 0.64 | 8 | 15 | 0.46 | 29 | 70 | 0.79 | 2 | 2 | 0.66 | 7 | 13 | 0.46 | 29 | 70 |
| CausalTab | 0.97 | 0 | 0 | 0.85 | 5 | 7 | 0.83 | 11 | 27 | 0.85 | 0 | 0 | 0.95 | 2 | 1 | 0.85 | 11 | 18 |

*Table 11.* Detailed results for `gp_simple` across different graph sizes ($d$) and metrics.

| Method | Observation | | | | | | | | | Mixed-intervention | | | | | | | | |
|---|---|---|---|---|---|---|---|---|---|---|---|---|---|---|---|---|---|---|
| | $d=5$ | | | $d=10$ | | | $d=20$ | | | $d=5$ | | | $d=10$ | | | $d=20$ | | |
| | F1 | SHD | SID | F1 | SHD | SID | F1 | SHD | SID | F1 | SHD | SID | F1 | SHD | SID | F1 | SHD | SID |
| RandomRegress | 0.34 | 7 | 9 | 0.26 | 34 | 42 | 0.18 | 151 | 179 | – | – | – | – | – | – | – | – | – |
| DAS | 0.84 | 1 | 2 | 0.72 | 9 | 10 | 0.61 | 35 | 46 | – | – | – | – | – | – | – | – | – |
| LiNGAM | 0.26 | 4 | 6 | 0.30 | 14 | 23 | 0.32 | 40 | 78 | – | – | – | – | – | – | – | – | – |
| PC | 0.52 | 3 | 5 | 0.46 | 11 | 24 | 0.42 | 33 | 91 | – | – | – | – | – | – | – | – | – |
| CDIS | 0.40 | 4 | 5 | 0.44 | 12 | 26 | 0.38 | 36 | 105 | 0.57 | 3 | 5 | 0.50 | 11 | 22 | 0.40 | 34 | 97 |
| GIES | 0.57 | 3 | 5 | 0.56 | 13 | 23 | 0.45 | 52 | 107 | 0.64 | 3 | 4 | 0.48 | 18 | 25 | 0.38 | 64 | 123 |
| IGSP | 0.58 | 3 | 5 | 0.48 | 13 | 27 | 0.31 | 56 | 140 | 0.55 | 3 | 5 | 0.47 | 13 | 28 | 0.32 | 53 | 135 |
| NOTEARS | 0.72 | 2 | 2 | 0.69 | 8 | 11 | 0.60 | 31 | 43 | – | – | – | – | – | – | – | – | – |
| NOTEARS-MLP | 0.83 | 1 | 1 | 0.79 | 6 | 8 | 0.70 | 31 | 31 | – | – | – | – | – | – | – | – | – |
| NoDAGS | – | – | – | – | – | – | – | – | – | 0.72 | 2 | 2 | 0.58 | 10 | 15 | 0.33 | 34 | 77 |
| DCDI | 0.50 | 1 | 2 | 0.52 | 13 | 58 | – | – | – | 0.22 | 7 | 6 | 0.56 | 17 | 36 | – | – | – |
| AVICI | 0.81 | 1 | 1 | 0.84 | 4 | 5 | 0.80 | 14 | 19 | 0.97 | 0 | 0 | 0.95 | 2 | 0 | 0.90 | 8 | 4 |
| CausalTab | 0.79 | 2 | 3 | 0.89 | 3 | 3 | 0.88 | 8 | 21 | 0.98 | 0 | 0 | 0.95 | 2 | 1 | 0.91 | 7 | 2 |

*Table 12.* Detailed results for `linear_gauss` across different graph sizes ($d$) and metrics.

| Method | Observation | | | | | | | | | Mixed-intervention | | | | | | | | |
|---|---|---|---|---|---|---|---|---|---|---|---|---|---|---|---|---|---|---|
| | $d=5$ | | | $d=10$ | | | $d=20$ | | | $d=5$ | | | $d=10$ | | | $d=20$ | | |
| | F1 | SHD | SID | F1 | SHD | SID | F1 | SHD | SID | F1 | SHD | SID | F1 | SHD | SID | F1 | SHD | SID |
| RandomRegress | 0.35 | 7 | 9 | 0.25 | 35 | 44 | 0.18 | 152 | 182 | – | – | – | – | – | – | – | – | – |
| DAS | 0.86 | 1 | 2 | 0.79 | 6 | 11 | 0.64 | 29 | 53 | – | – | – | – | – | – | – | – | – |
| LiNGAM | 0.47 | 4 | 5 | 0.46 | 13 | 20 | 0.42 | 41 | 75 | – | – | – | – | – | – | – | – | – |
| PC | 0.67 | 2 | 4 | 0.54 | 10 | 21 | 0.45 | 30 | 88 | – | – | – | – | – | – | – | – | – |
| CDIS | 0.56 | 3 | 4 | 0.49 | 11 | 23 | 0.42 | 32 | 92 | 0.71 | 2 | 3 | 0.52 | 10 | 24 | 0.41 | 33 | 94 |
| GIES | 0.69 | 2 | 4 | 0.68 | 8 | 19 | 0.61 | 36 | 89 | 0.98 | 0 | 0 | 0.94 | 2 | 3 | 0.88 | 11 | 27 |
| IGSP | 0.75 | 1 | 4 | 0.72 | 6 | 17 | 0.64 | 28 | 78 | 0.71 | 2 | 4 | 0.71 | 7 | 17 | 0.58 | 33 | 93 |
| NOTEARS | 0.87 | 1 | 1 | 0.85 | 4 | 5 | 0.85 | 11 | 19 | – | – | – | – | – | – | – | – | – |
| NOTEARS-MLP | 0.85 | 1 | 2 | 0.67 | 8 | 13 | 0.58 | 25 | 62 | – | – | – | – | – | – | – | – | – |
| NoDAGS | – | – | – | – | – | – | – | – | – | 0.95 | 1 | 0 | 0.69 | 8 | 10 | 0.19 | 37 | 84 |
| DCDI | 0.27 | 8 | 13 | 0.42 | 27 | 53 | – | – | – | 0.44 | 5 | 5 | 0.50 | 14 | 36 | – | – | – |
| AVICI | 0.58 | 3 | 5 | 0.62 | 10 | 14 | 0.56 | 28 | 53 | 0.98 | 0 | 0 | 0.93 | 2 | 0 | 0.84 | 13 | 5 |
| CausalTab | 0.69 | 3 | 5 | 0.72 | 8 | 13 | 0.81 | 13 | 25 | 1.00 | 0 | 0 | 0.99 | 0 | 0 | 0.96 | 3 | 1 |

*Table 13.* Detailed results for `linear_graph` across different graph sizes ($d$) and metrics.

| Method | Observation | | | | | | | | | Mixed-intervention | | | | | | | | |
|---|---|---|---|---|---|---|---|---|---|---|---|---|---|---|---|---|---|---|
| | $d = 5$ | | | $d = 10$ | | | $d = 20$ | | | $d = 5$ | | | $d = 10$ | | | $d = 20$ | | |
| | F1 | SHD | SID | F1 | SHD | SID | F1 | SHD | SID | F1 | SHD | SID | F1 | SHD | SID | F1 | SHD | SID |
| RandomRegress | 0.26 | 7 | 8 | 0.22 | 35 | 41 | 0.17 | 153 | 179 | – | – | – | – | – | – | – | – | – |
| DAS | 0.76 | 1 | 2 | 0.78 | 5 | 8 | 0.64 | 26 | 48 | – | – | – | – | – | – | – | – | – |
| LiNGAM | 0.44 | 2 | 3 | 0.46 | 10 | 14 | 0.42 | 39 | 67 | – | – | – | – | – | – | – | – | – |
| PC | 0.56 | 1 | 2 | 0.60 | 6 | 13 | 0.45 | 28 | 74 | – | – | – | – | – | – | – | – | – |
| CDIS | 0.30 | 2 | 2 | 0.52 | 7 | 13 | 0.44 | 29 | 73 | 0.65 | 1 | 1 | 0.62 | 6 | 11 | 0.45 | 28 | 71 |
| GIES | 0.59 | 1 | 2 | 0.73 | 5 | 10 | 0.67 | 25 | 59 | 0.91 | 0 | 0 | 0.96 | 1 | 2 | 0.89 | 9 | 24 |
| IGSP | 0.57 | 1 | 2 | 0.68 | 5 | 12 | 0.52 | 38 | 98 | 0.65 | 1 | 2 | 0.70 | 5 | 11 | 0.53 | 36 | 92 |
| NOTEARS | 0.85 | 0 | 0 | 0.87 | 3 | 3 | 0.86 | 9 | 15 | – | – | – | – | – | – | – | – | – |
| NOTEARS-MLP | 0.86 | 0 | 1 | 0.73 | 6 | 9 | 0.55 | 25 | 54 | – | – | – | – | – | – | – | – | – |
| NoDAGS | – | – | – | – | – | – | – | – | – | 0.91 | 0 | 0 | 0.79 | 5 | 5 | 0.26 | 33 | 63 |
| DCDI | 0.67 | 2 | 2 | 0.31 | 31 | 46 | – | – | – | 0.00 | 2 | 3 | 0.50 | 13 | 62 | – | – | – |
| AVICI | 0.49 | 1 | 2 | 0.64 | 6 | 8 | 0.59 | 25 | 40 | 0.94 | 0 | 0 | 0.96 | 1 | 0 | 0.85 | 12 | 3 |
| CausalTab | 0.56 | 1 | 2 | 0.69 | 5 | 7 | 0.80 | 13 | 16 | 1.00 | 0 | 0 | 0.99 | 0 | 0 | 0.96 | 3 | 2 |

*Table 14.* Detailed results for `linear_nongauss` across different graph sizes ($d$) and metrics.

| Method | Observation | | | | | | | | | Mixed-intervention | | | | | | | | |
|---|---|---|---|---|---|---|---|---|---|---|---|---|---|---|---|---|---|---|
| | $d = 5$ | | | $d = 10$ | | | $d = 20$ | | | $d = 5$ | | | $d = 10$ | | | $d = 20$ | | |
| | F1 | SHD | SID | F1 | SHD | SID | F1 | SHD | SID | F1 | SHD | SID | F1 | SHD | SID | F1 | SHD | SID |
| RandomRegress | 0.32 | 7 | 10 | 0.28 | 34 | 43 | 0.18 | 153 | 182 | – | – | – | – | – | – | – | – | – |
| DAS | 0.94 | 1 | 1 | 0.84 | 5 | 7 | 0.69 | 26 | 44 | – | – | – | – | – | – | – | – | – |
| LiNGAM | 0.99 | 0 | 0 | 0.99 | 0 | 0 | 0.99 | 1 | 1 | – | – | – | – | – | – | – | – | – |
| PC | 0.64 | 2 | 5 | 0.54 | 10 | 22 | 0.43 | 31 | 90 | – | – | – | – | – | – | – | – | – |
| CDIS | 0.47 | 3 | 5 | 0.52 | 11 | 22 | 0.41 | 33 | 99 | 0.66 | 2 | 4 | 0.58 | 9 | 20 | 0.42 | 32 | 95 |
| GIES | 0.69 | 2 | 5 | 0.67 | 9 | 20 | 0.58 | 40 | 94 | 0.98 | 0 | 0 | 0.95 | 1 | 3 | 0.91 | 8 | 22 |
| IGSP | 0.72 | 2 | 4 | 0.73 | 6 | 17 | 0.63 | 30 | 82 | 0.71 | 2 | 4 | 0.73 | 6 | 15 | 0.62 | 30 | 83 |
| NOTEARS | 0.85 | 1 | 1 | 0.83 | 4 | 6 | 0.85 | 11 | 19 | – | – | – | – | – | – | – | – | – |
| NOTEARS-MLP | 0.82 | 2 | 2 | 0.67 | 8 | 14 | 0.57 | 27 | 63 | – | – | – | – | – | – | – | – | – |
| NoDAGS | – | – | – | – | – | – | – | – | – | 0.92 | 1 | 1 | 0.71 | 7 | 9 | 0.21 | 36 | 79 |
| DCDI | 0.29 | 7 | 14 | 0.42 | 32 | 59 | – | – | – | 0.73 | 3 | 3 | 0.52 | 21 | 64 | – | – | – |
| AVICI | 0.28 | 5 | 9 | 0.41 | 13 | 26 | 0.39 | 34 | 79 | 0.96 | 0 | 0 | 0.92 | 3 | 1 | 0.85 | 12 | 7 |
| CausalTab | 0.99 | 0 | 0 | 0.94 | 2 | 0 | 0.92 | 6 | 4 | 1.00 | 0 | 0 | 0.98 | 1 | 0 | 0.96 | 4 | 0 |

*Table 15.* Detailed results for `mul_noise` across different graph sizes ($d$) and metrics.

| Method | Observation | | | | | | | | | Mixed-intervention | | | | | | | | |
|---|---|---|---|---|---|---|---|---|---|---|---|---|---|---|---|---|---|---|
| | $d = 5$ | | | $d = 10$ | | | $d = 20$ | | | $d = 5$ | | | $d = 10$ | | | $d = 20$ | | |
| | F1 | SHD | SID | F1 | SHD | SID | F1 | SHD | SID | F1 | SHD | SID | F1 | SHD | SID | F1 | SHD | SID |
| RandomRegress | 0.33 | 7 | 9 | 0.26 | 34 | 42 | 0.18 | 150 | 177 | – | – | – | – | – | – | – | – | – |
| DAS | 0.82 | 1 | 2 | 0.74 | 7 | 10 | 0.64 | 28 | 46 | – | – | – | – | – | – | – | – | – |
| LiNGAM | 0.58 | 3 | 4 | 0.51 | 11 | 16 | 0.46 | 35 | 62 | – | – | – | – | – | – | – | – | – |
| PC | 0.60 | 2 | 5 | 0.53 | 9 | 22 | 0.48 | 27 | 90 | – | – | – | – | – | – | – | – | – |
| CDIS | 0.49 | 3 | 5 | 0.51 | 9 | 21 | 0.46 | 28 | 93 | 0.60 | 2 | 5 | 0.54 | 9 | 22 | 0.46 | 29 | 96 |
| GIES | 0.65 | 2 | 5 | 0.64 | 8 | 20 | 0.65 | 25 | 83 | 0.96 | 0 | 0 | 0.92 | 3 | 2 | 0.83 | 16 | 28 |
| IGSP | 0.63 | 2 | 5 | 0.61 | 8 | 20 | 0.54 | 28 | 95 | 0.51 | 3 | 6 | 0.58 | 9 | 22 | 0.49 | 32 | 106 |
| NOTEARS | 0.16 | 5 | 6 | 0.31 | 12 | 18 | 0.43 | 30 | 64 | – | – | – | – | – | – | – | – | – |
| NOTEARS-MLP | 0.35 | 4 | 5 | 0.47 | 11 | 16 | 0.54 | 31 | 57 | – | – | – | – | – | – | – | – | – |
| NoDAGS | – | – | – | – | – | – | – | – | – | 0.92 | 1 | 0 | 0.78 | 6 | 6 | 0.41 | 31 | 70 |
| DCDI | 0.40 | 8 | 12 | 0.51 | 22 | 57 | – | – | – | 0.43 | 7 | 11 | 0.38 | 30 | 51 | – | – | – |
| AVICI | 0.50 | 3 | 6 | 0.51 | 10 | 18 | 0.40 | 32 | 80 | 0.96 | 0 | 0 | 0.91 | 3 | 0 | 0.81 | 15 | 7 |
| CausalTab | 1.00 | 0 | 0 | 0.95 | 1 | 1 | 0.87 | 10 | 6 | 0.98 | 0 | 0 | 0.99 | 0 | 0 | 0.94 | 6 | 0 |

*Table 16.* Detailed results for `pfn` across different graph sizes (*d*) and metrics.

| Method | Observation | | | | | | | | | Mixed-intervention | | | | | | | | |
| --- | --- | --- | --- | --- | --- | --- | --- | --- | --- | --- | --- | --- | --- | --- | --- | --- | --- | --- |
| | $d = 5$ | | | $d = 10$ | | | $d = 20$ | | | $d = 5$ | | | $d = 10$ | | | $d = 20$ | | |
| | F1 | SHD | SID | F1 | SHD | SID | F1 | SHD | SID | F1 | SHD | SID | F1 | SHD | SID | F1 | SHD | SID |
| RandomRegress | 0.26 | 8 | 10 | 0.18 | 35 | 44 | 0.14 | 153 | 182 | – | – | – | – | – | – | – | – | – |
| DAS | 0.33 | 5 | 8 | 0.25 | 14 | 35 | 0.20 | 49 | 138 | – | – | – | – | – | – | – | – | – |
| LiNGAM | 0.14 | 4 | 8 | 0.12 | 13 | 31 | 0.16 | 39 | 111 | – | – | – | – | – | – | – | – | – |
| PC | 0.49 | 3 | 6 | 0.44 | 9 | 27 | 0.32 | 32 | 112 | – | – | – | – | – | – | – | – | – |
| CDIS | 0.38 | 3 | 6 | 0.36 | 10 | 29 | 0.33 | 31 | 110 | 0.42 | 3 | 6 | 0.40 | 9 | 28 | 0.32 | 30 | 106 |
| GIES | 0.61 | 2 | 4 | 0.55 | 8 | 21 | 0.51 | 31 | 84 | 0.80 | 2 | 1 | 0.74 | 6 | 11 | 0.67 | 25 | 50 |
| IGSP | 0.52 | 3 | 5 | 0.44 | 9 | 24 | 0.35 | 37 | 113 | 0.42 | 3 | 6 | 0.45 | 9 | 25 | 0.28 | 31 | 100 |
| NOTEARS | 0.22 | 4 | 7 | 0.18 | 11 | 28 | 0.18 | 34 | 104 | – | – | – | – | – | – | – | – | – |
| NOTEARS-MLP | 0.24 | 4 | 7 | 0.21 | 13 | 30 | 0.15 | 38 | 110 | – | – | – | – | – | – | – | – | – |
| NoDAGS | – | – | – | – | – | – | – | – | – | 0.52 | 3 | 5 | 0.42 | 8 | 24 | 0.24 | 28 | 93 |
| DCDI | 0.44 | 5 | 10 | 0.27 | 30 | 57 | – | – | – | 0.75 | 4 | 8 | 0.32 | 31 | 51 | – | – | – |
| AVICI | 0.44 | 3 | 5 | 0.41 | 10 | 24 | 0.36 | 29 | 96 | 0.78 | 1 | 2 | 0.69 | 5 | 15 | 0.62 | 17 | 66 |
| CausalTab | 0.63 | 2 | 4 | 0.81 | 3 | 16 | 0.70 | 12 | 73 | 0.96 | 0 | 1 | 0.83 | 2 | 8 | 0.93 | 2 | 8 |

