# OpenReview forum: "CausalTab: Pretraining Across Causal Environments for Tabular Causal Discovery"
_ICML.cc/2026/Workshop/FMSD — FMSD @ ICML 2026 Poster_

### Official Review · Reviewer_9t48 · 2026-05-19
**CausalTab**

**Rating:** 6
**Confidence:** 4

**Review:**

## Summary

This paper proposes CausalTab, a causal discovery foundation model (CDFM) that takes a tabular dataset (values $\mathbf{X}^{\text{val}} \in \mathbb{R}^{N \times d}$ plus optional intervention masks $\mathbf{M} \in \{0,1\}^{N \times d}$) and predicts directed edge probabilities $\hat{\mathbf{P}} \in (0,1)^{d \times d}$ in a single forward pass. The architecture uses axial attention (variables × samples), max-pooling over samples to produce variable tokens, and asymmetric parent/child projections with biaffine scoring for edge prediction.

## Strengths

- **Clear thesis well-executed**: The argument that environment breadth (graphs × mechanisms × noise × dimensions × regimes) is the key design axis is articulated upfront and supported by the prior-strength ablation (Table 6: SHD 26.0 → 3.6, F1 0.659 → 0.888 from weak to strong prior). This is a meaningful contribution to how CDFMs should be pretrained.
- **Comprehensive benchmarking**: 13 baselines spanning constraint-based (PC), score-based (GIES), continuous-optimization (NOTEARS variants), functional (LiNGAM), amortized (AVICI, DCDI), cyclic (NoDAGS), and regression controls (DAS, RandomRegress). Few CDFM papers cover this breadth.
- **Mixed-interventional support**: The intervention mask handling is principled and the empirical gains under mixed evidence (Table 1: Synthetic Mixed F1: 0.952 vs. AVICI 0.844 and GIES 0.780) are substantial.
- **Semantic benchmark is a thoughtful complement**: While clearly synthetic and SCM-grounded (as the authors honestly state), the authored variable semantics enable interpretable probes that anonymous benchmarks cannot. The PCA visualization for the permit-backlog scenario is a nice qualitative addition.
- **Embedding analysis (Appendix H.2)**: The probing experiments showing pooled embeddings can predict graph statistics (edge count, avg degree, max in-degree, DAG depth) with $R^2 \geq 0.716$ and 30-62% lower MAE than raw features is genuinely interesting evidence that learned representations encode structural information.
- **Computational efficiency analysis (Figure 9)**: Concrete wall-clock comparisons (CausalTab 0.9s/sample vs. several baselines > 120s) make the practical case for amortization.
- **Honest limitations**: The authors explicitly note that even the "semantic" benchmark is simulator-grounded rather than field telemetry, and that observation-only performance is naturally more ambiguous. They also acknowledge dimension as the primary bottleneck (Appendix H.4).
- **High-dim scalability analysis**: Appendix D.2 reports results up to $d = 300$ on observational data, which is genuinely useful for understanding amortization limits.

## Areas for Improvement

- **Limited methodological novelty**: Axial attention is from Ho et al. 2019, biaffine scoring from Dozat & Manning 2017, asymmetric edge heads from Lorch et al. 2022 (AVICI). The architecture is competently composed but not new. The contribution is empirical, which is fine, but the framing as a method paper undersells this.
- **Observation-only gains are tighter**: Under purely observational data, CausalTab's rank advantage shrinks substantially and specialized methods (e.g., DAS, GIES) can match or exceed it on specific families. This is honestly reported but is the regime that matters most when interventions are unavailable, which is common in practice. The paper would benefit from more analysis of why this gap is smaller and what to do about it.
- **Semantic benchmark validity concerns**: The benchmark uses an LLM-aided audit (Appendix G.13), which the authors honestly acknowledge as "scalable documentation and consistency support — not exhaustive semantic certification." But this still leaves open whether the semantic SCMs are genuinely harder/different than anonymous ones or just renamed. The mixed-regime semantic F1 of 0.971 is very high — does this suggest the semantic benchmark might actually be easier?
- **Threshold-dependent results**: F1 and SHD depend on edge-probability thresholds. The "default decoding rule consistent with the evaluation pipeline" (Appendix I) is opaque. Sensitivity to thresholds should be reported, especially for AUROC-style measures.
- **"Pretraining task diversity drives generalization" — needs more careful ablation**: Table 6 compares weak vs. strong prior, but the strong prior changes many things at once (graphs, mechanisms, noise, structured templates, OOD neural mechanisms). Isolating which components matter most would substantially strengthen the central claim.
- **No comparison to recent CDFMs beyond AVICI**: Montagna et al. (2024) is cited but not compared empirically. Other recent CDFMs may also be relevant.
- **Real-world data is entirely absent**: Despite the focus on causal discovery, no real (non-authored) dataset is used. Even one standard benchmark (e.g., Protein-signaling by Sachs et al. 2005, Krebs cycle metabolic data from He et al. 2026) would strengthen the contribution.
- **Cycle pruning is mentioned but not characterized**: The "optional cycle pruning runs only at inference" — how often does it activate, and how does it affect metrics?


## Detailed Comments

1. The axial attention design with max-pooling over $N$ to obtain variable tokens loses information about sample-level variation. Have you tried mean-pooling or attention pooling? Sample heterogeneity could be informative for causal direction (e.g., variance differences under interventions).
2. The "structurally intervened" mask $\mathbf{M}$ is per-row/per-column. Does the model actually use both pieces of information (which row, which column), or is the row-level "intervened or not" signal sufficient? An ablation would clarify.
3. The motif-tag analysis (Table 4) is interesting but the table is dense and hard to read. Consider visualizing as a heatmap.
4. The pfn family is OOD by construction. Are there other ways CausalTab fails (e.g., extreme heterogeneity, very dense graphs)?
5. For high-d scalability (Figure 5), CausalTab is best at $d=50, 100, 300$ on `gp_hard_obs` and `pfn_obs`. But some baselines drop out due to timeout — is this a fair comparison? Could you report results at fixed compute budget?
6. The "semantic causal environment benchmark" — please clarify if the JSON specifications and generation code will be released. This determines whether the benchmark can be used by others.
7. The LLM-aided audit (Appendix G.13) should specify the LLM version, prompts, and inter-rater reliability if multiple passes were made.

## Justification of Score

This is a well-executed empirical paper with a clear thesis (environment-diversity > architecture tweaks for CDFMs) and broad evaluation. The contribution is more about pretraining methodology and benchmarking than methodological novelty, which is fine for a workshop focused on foundation models for structured data. However, the lack of real-world validation, the observation-only gains being modest, and the entanglement of multiple changes in the prior-strength ablation are non-trivial limitations.

---

### Official Review · Reviewer_SBGP · 2026-05-21
**Good mixed-intervention results**

**Rating:** 6
**Confidence:** 3

**Review:**

Summary:
The paper proposes CausalTab, a tabular causal discovery foundation model. It uses axial attention over samples and variables, takes intervention masks as input, and predicts directed edges. The main idea is broad causal pretraining across different graph priors, mechanisms, noise types, dimensions, and intervention regimes.

Strengths:
The paper fits the workshop well. The mixed-interventional results are strong, and CausalTab has the best synthetic mixed rank/F1 and semantic mixed F1 in the main table. The comparison with classical and amortized baselines is also useful.

Areas for Improvement:
The observation-only results are less clearly dominant. In semantic observation-only, GIES slightly beats CausalTab on F1. The claims should focus more on the mixed-interventional setting.

Detailed Comments:
The semantic benchmark and PCA plot are interesting, but they are still simulator-grounded and qualitative. I would avoid overinterpreting the PCA plot as evidence of semantic causal understanding.

Justification of Score:
Good workshop paper. The main results are useful, especially with interventions. My concerns are mostly about claim scope and the semantic evaluation.